# Glycine inhibits NINJ1 membrane clustering to suppress plasma membrane rupture in cell death

Jazlyn P Borges[1†], Ragnhild SR Sætra[2†], Allen Volchuk[3], Marit Bugge[2], Pascal Devant[4], Bjørnar Sporsheim[2], Bridget R Kilburn[1], Charles L Evavold[5], Jonathan C Kagan[4], Neil M Goldenberg[3,6,7], Trude Helen Flo[2,4], Benjamin Ethan Steinberg[1,6,7]*

[1]Program in Neuroscience and Mental Health, Hospital for Sick Children, Toronto, Canada; [2]Centre of Molecular Inflammation Research, Department of Clinical and Molecular Medicine, Faculty of Medicine and Health Sciences, Norwegian University of Science and Technology, Trondheim, Norway; [3]Program in Cell Biology, Hospital for Sick Children, Toronto, Canada; [4]Division of Gastroenterology, Boston Children's Hospital and Harvard Medical School, Boston, United States; [5]Ragon Institute of MGH, MIT and Harvard, Boston, United States; [6]Department of Anesthesia and Pain Medicine, Hospital for Sick Children, Toronto, Canada; [7]Department of Anesthesiology and Pain Medicine, University of Toronto, Toronto, Canada

**\*For correspondence:**
benjamin.steinberg@sickkids.ca

[†]These authors contributed equally to this work

**Competing interest:** The authors declare that no competing interests exist.

**Abstract** First recognized more than 30 years ago, glycine protects cells against rupture from diverse types of injury. This robust and widely observed effect has been speculated to target a late downstream process common to multiple modes of tissue injury. The molecular target of glycine that mediates cytoprotection, however, remains elusive. Here, we show that glycine works at the level of NINJ1, a newly identified executioner of plasma membrane rupture in pyroptosis, necrosis, and post-apoptosis lysis. NINJ1 is thought to cluster within the plasma membrane to cause cell rupture. We demonstrate that the execution of pyroptotic cell rupture is similar for human and mouse NINJ1 and that NINJ1 knockout functionally and morphologically phenocopies glycine cytoprotection in macrophages undergoing lytic cell death. Next, we show that glycine prevents NINJ1 clustering by either direct or indirect mechanisms. In pyroptosis, glycine preserves cellular integrity but does not affect upstream inflammasome activities or accompanying energetic cell death. By positioning NINJ1 clustering as a glycine target, our data resolve a long-standing mechanism for glycine-mediated cytoprotection. This new understanding will inform the development of cell preservation strategies to counter pathologic lytic cell death.

## Editor's evaluation

It's been widely known that the amino acid Glycine can work as a cytoprotectant and inhibit cell death-associated plasma membrane rupture. However, a long-standing question has been: how does Glycine cytoprotection work? In this manuscript, the authors demonstrate that Glycine treatment inhibits the clustering of NINJ1 to preserve membrane integrity, which provides a significant advance in the cell death field.

## Introduction

The amino acid glycine has long been known to protect cells against plasma membrane rupture induced by a diverse set of injurious stimuli. This cytoprotective effect was first described in 1987, where glycine was found to protect renal tubular cells against hypoxic injury (*Weinberg et al., 1987*). Since, similar observations have been replicated across immune, parenchymal, endothelial, and neuronal cell types in multiple injury models (reviewed in *Weinberg et al., 2016*). These include the lytic cell death pathways pyroptosis (*Loomis et al., 2019*; *Volchuk et al., 2020*) and necrosis (*Estacion et al., 2003*).

While the robust cytoprotective effects of glycine are widespread, the molecular target and mechanism by which glycine protects cells and tissues remain elusive. Many cellular processes have been investigated but ultimately ruled out as underlying this phenomenon (*Weinberg et al., 2016*). These include glycine or ATP metabolic pathways (*Weinberg et al., 1991*), intracellular calcium or pH regulation (*Weinberg et al., 1994*), cytoskeleton stabilization (*Chen et al., 1997*), and chloride conductance (*Loomis et al., 2019*). It has been speculated that glycine targets a membrane receptor but is unlikely to involve canonical glycine receptors (*Loomis et al., 2019*; *Weinberg et al., 2016*). Lastly, as a cytoprotective agent, glycine has oftentimes been described as an osmoprotectant (*Fink and Cookson, 2006*). However, its small size should generally allow free passage through the large membrane conduits involved in lytic cell death pathways, such as the gasdermin D pores formed during pyroptosis that are known to allow transport of small proteins and ions across lipid bilayers (*Evavold et al., 2018*; *Heilig et al., 2018*; *Xia et al., 2021*). The free permeability of the plasma membrane to glycine through large conduits or amino acid transporters would negate a protective osmotic effect. Moreover, it remains unknown whether glycine's cytoprotective effect is through an extracellular or intracellular mode of action and, therefore, whether glycine's entry into the cell is required for cytoprotection. A mechanistic knowledge of how glycine preserves cellular integrity would inform the development of cell and tissue preservation strategies. This mechanistic understanding of glycine cytoprotection has clear clinical implications, such as for limiting inflammation in ischemia-reperfusion injuries where multiple lytic cell death pathways are activated to cause tissue damage and morbidity (*Del Re et al., 2019*).

Given how widely observed the phenomenon is, it stands to reason that glycine targets a late downstream process common to multiple tissue injury models. Notably, the transmembrane protein ninjurin-1 (NINJ1) was recently identified as the common executioner of plasma membrane rupture in pyroptosis, necrosis, and post-apoptosis lysis (*Kayagaki et al., 2021*). Post-apoptosis lysis refers to the cell membrane rupture that occurs when phagocytes are unable to scavenge apoptotic cells that have otherwise completed the apoptotic program (*Silva, 2010*). NINJ1 is a 16 kilodalton protein with two transmembrane regions that reside in the plasma membrane with prior reports, suggesting a function in cell adhesion (*Araki et al., 1997*; *Araki and Milbrandt, 1996*). NINJ1 has been implicated in multiple inflammatory conditions, such as nerve injury, atherosclerosis, ischemic brain injury, and tumorigenesis (*Araki and Milbrandt, 1996*; *Jeon et al., 2020*; *Kim et al., 2020*; *Yang et al., 2017*). The mechanism by which NINJ1 mediates lytic cell death coincides with its clustering within the plasma membrane, and this multimerization is required for membrane disruption by an unknown mechanism (*Kayagaki et al., 2021*).

Here, we hypothesized that glycine targets NINJ1 to mediate its cytoprotective effect. We first demonstrate that NINJ1 knockout or silencing functionally and morphologically phenocopies glycine cytoprotection in mouse and human primary macrophages stimulated to undergo various forms of lytic cell death. Next, we show that glycine treatment prevents NINJ1 clustering within the plasma membrane, thereby preserving its integrity without affecting upstream programmed cell death signaling. By identifying NINJ1-dependent plasma membrane rupture as a glycine target, our data help resolve a long-standing mechanism of glycine cytoprotection.

## Results

### NINJ1 deficiency functionally and morphologically phenocopies glycine cytoprotection

The published literature suggests a potential association between NINJ1 and glycine given that NINJ1 knockout and glycine treatment protect cells against a common set of lytic cell death pathways.

Moreover, in pyroptosis, both allow for IL-1β secretion through the gasdermin D pore while preventing final membrane lysis (*Kayagaki et al., 2021*; *Volchuk et al., 2020*). We first sought to extend these associations by further characterizing functional and morphological similarities between NINJ1 knockout and glycine cytoprotection in pyroptosis, necrosis, and post-apoptosis lysis.

Using a CRISPR-Cas9 system, we generated a NINJ1 knockout cell line in mouse immortalized bone marrow-derived macrophages (iBMDMs; *Figure 1A*). Pyroptosis was induced in LPS(lipopolysaccharide)-primed wildtype and NINJ1 knockout iBMDM with nigericin (20 µM, 2 hr) in the absence or presence of 5 mM glycine. Glycine treatment at 5 mM was based on the reported literature (*Loomis et al., 2019*; *Volchuk et al., 2020*) and a dose-finding study (*Figure 1—figure supplement 1*). We performed a colorimetric assay to test for LDH release, a common marker of cell rupture. Both NINJ1 knockout and glycine treatment in wildtype iBMDM protected against cytotoxicity (*Figure 1B*). Similarly, cell membrane integrity was preserved following apoptosis (*Figure 1C*) and necrosis (*Figure 1D*) induced with venetoclax (25 µM, 16 hr) and pneumolysin (0.5 µg/mL, 15 min), respectively. Importantly, in NINJ1 knockout macrophages, treatment with glycine during lytic cell death did not confer any additional protection against cell lysis, suggesting that these distinct perturbations affect the same process (*Figure 1B–D*).

When visualizing glycine-treated pyroptotic macrophages, we observed prominent plasma membrane ballooning (*Figure 1E*). This impressive morphology was comparable to that of LPS-primed NINJ1 knockout macrophages stimulated to undergo pyroptosis with nigericin, which we (*Figure 1E*) and others (*Kayagaki et al., 2021*) have observed. Together, these functional and morphologic similarities buttress the association between glycine cytoprotection and NINJ1-mediated plasma membrane rupture.

To further corroborate the above findings, we conducted pyroptosis assays in RAW mouse macrophages reconstituted with the inflammasome adaptor ASC (hereafter referred to as RAW-ASC). NINJ1 protein expression was ablated through CRISPR-Cas9 targeting as compared with the parental line (*Figure 1—figure supplement 2A*). Pyroptosis stimulation in this orthogonal murine macrophage model demonstrates NINJ1 deficiency functionally (*Figure 1—figure supplement 2B*) and morphologically (*Figure 1—figure supplement 2C*) phenocopies glycine treatment as was seen in the iBMDM system.

Human NINJ1 is 90% homologous to the mouse protein and was shown to be toxic when overexpressed in HEK293T cells (*Kayagaki et al., 2021*), but it is not known if endogenous NINJ1 is activated and mediates plasma membrane rupture in human cells undergoing programmed cell death. We knocked down NINJ1 in primary human monocyte-derived macrophages (hMDMs) using siRNA specific to NINJ1 (siNINJ1) and compared to non-targeting control siRNA (siCtrl) with an efficiency approaching 70% (*Figure 4—figure supplement 2A*). Both NINJ1 knockdown and glycine (50 mM; *Figure 2—figure supplement 1*) independently and completely prevented pyroptosis (LDH release) induced by nigericin (20 µM, 2 hr) in LPS-primed hMDMs. Notably, again there was no additional effect with combined NINJ1 deficiency and glycine treatment arguing that NINJ1 and glycine are involved in the same process in primary human cell death pathways (*Figure 2A*). In contrast, human macrophages induced to undergo necroptosis (zVAD 20 µM + SMAC mimetic BV6 10 µM followed by TNF 10 ng/mL) were unprotected by glycine treatment (*Figure 2—figure supplement 2*), consistent with the observation that *Ninj1* knockout does not confer protection in necroptosis likely due to sufficient cytolytic activity of MLKL pores (*Kayagaki et al., 2021*). Like mouse macrophages, both siNINJ1 and glycine sustained the ballooned morphology of pyroptotic hMDMs (*Figure 2B*). While pyroptotic human macrophages maintained their plasma membrane integrity in the presence of glycine as demonstrated by suppressed LDH release, we next confirmed that they had nevertheless lost viability as measured by the loss of mitochondrial membrane potential and cellular ATP content (*Figure 2—figure supplement 3*).

Together, these data using macrophage models from mouse and human demonstrate that genetic deletion or silencing of NINJ1 functionally and morphologically parallels glycine-mediated cytoprotection.

## Glycine targets NINJ1 clustering to prevent membrane rupture

We therefore next posited that glycine targets NINJ1 as part of its cytoprotective effect. The working model of NINJ1-mediated membrane rupture involves its clustering within the plasma membrane

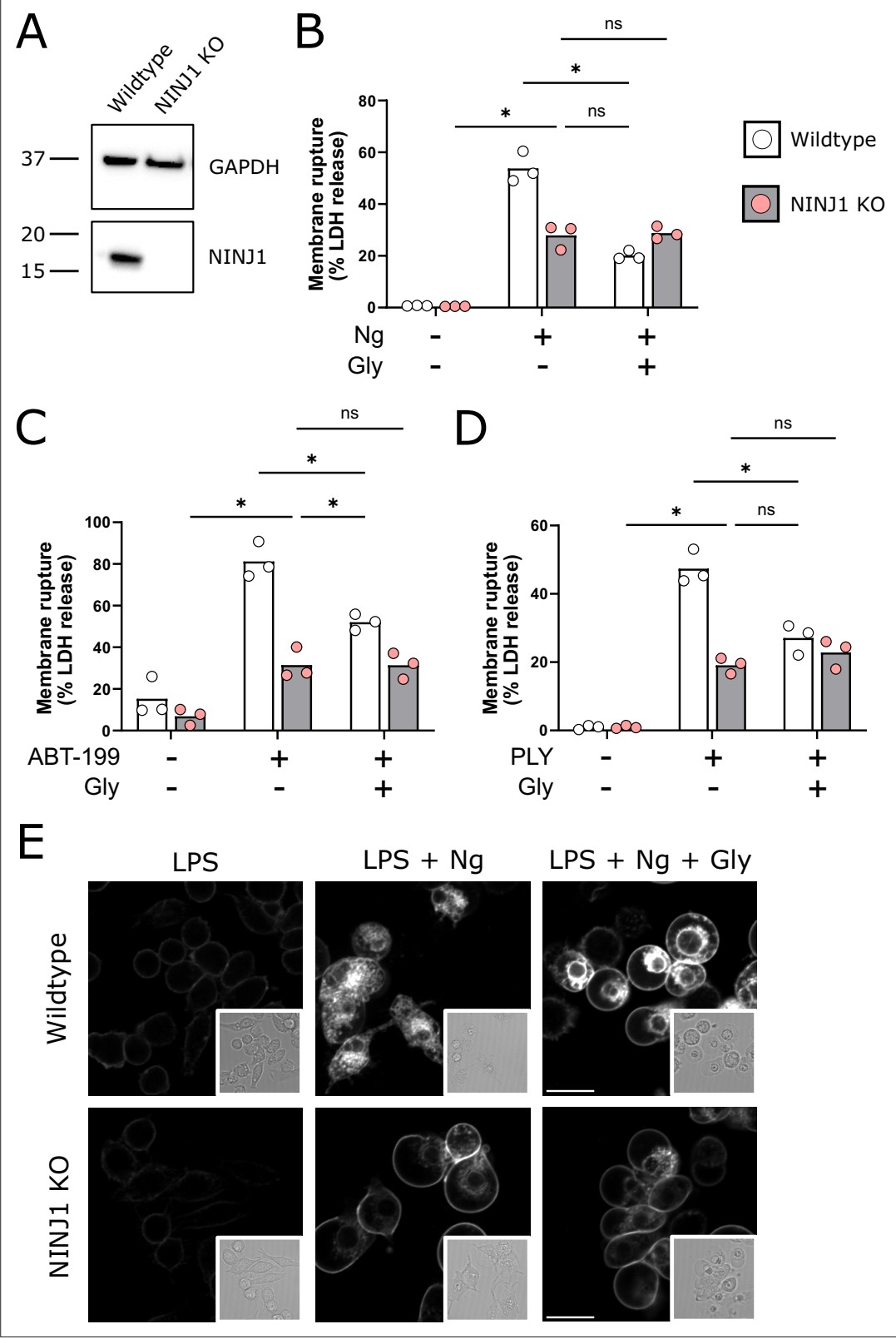

**Figure 1.** NINJ1 knockout functionally and morphologically phenocopies glycine cytoprotection. (**A**) Immunoblot analysis demonstrating NINJ1 knockout in immortalized bone marrow-derived macrophage (iBMDM). GAPDH(glyceraldehyde-3-phosphate dehydrogenase) is presented as a loading control. (**B–D**) Wildtype and NINJ1 knockout iBMDM were induced to undergo pyroptosis (LPS + nigericin), post-apoptosis lysis (secondary

*Figure 1 continued on next page*

*Figure 1 continued*

necrosis; venetoclax, ABT-199), or necrosis (pneumolysin) with or without 5 mM glycine treatment. Cytotoxicity was evaluated by measuring the levels of LDH in the supernatant. Cytotoxicity decreased in glycine-treated wildtype cells comparably to NINJ1 knockout across each of (**B**) pyroptosis, (**C**) post-apoptosis lysis, and (**D**) necrosis. Glycine treatment of NINJ1 KO cells provided no additional protection to knockout cells treated without glycine. Data are expressed as supernatant LDH as a % of total LDH from lysates and supernatants, from n = 3 independent experiments for each cell death type. Individual data points are shown along with their mean. * p<0.05 by ANOVA with Tukey's multiple comparison correction. (**E**) LPS-primed wildtype iBMDM induced to undergo pyroptosis in the presence of glycine demonstrate similar plasma membrane ballooning to NINJ1 knockout iBMDM induced to undergo pyroptosis. Membrane ballooning is shown in live cells labeled with the plasma membrane dye FM4-64. The corresponding brightfield image is shown in the inset. Scale bar 15 µm.

The online version of this article includes the following source data and figure supplement(s) for figure 1:

**Source data 1.** Numerical values for experimental data plotted in *Figure 1*.

**Figure supplement 1.** Glycine dose-dependently inhibits rupture of pyroptotic mouse macrophages with greater potency than structurally similar amino acids through inhibition of NINJ1 clustering.

**Figure supplement 1—source data 1.** Numerical values for experimental data plotted in *Figure 1—figure supplement 1*.

**Figure supplement 2.** Glycine cytoprotection phenocopies NINJ1 knockout and prevents NINJ1 aggregation in a mouse macrophage cell line.

**Figure supplement 2—source data 1.** Numerical values for experimental data plotted in *Figure 1—figure supplement 2*.

(*Kayagaki et al., 2021*). Point mutants (e.g. K45Q and A59P) that prevent membrane rupture also fail to cluster upon induction of pyroptosis (*Kayagaki et al., 2021*), further reinforcing a connection between the clustering process and the capacity of NINJ1 to rupture membranes. To establish whether glycine directly interferes with NINJ1-mediated cell death, we next evaluated the effect of glycine treatment on NINJ1 clustering during cell lysis in pyroptosis, post-apoptosis lysis, and necrosis.

Using a biochemical native-PAGE approach that maintains native protein interactions, endogenous NINJ1 of otherwise untreated primary mouse bone marrow-derived macrophages (BMDM) migrates at approximately 40 kDa, potentially indicative of NINJ1 dimers or trimers in unstimulated cells (*Figure 3A*) and consistent with its known ability to form homotypic interactions (*Bae et al., 2017*). By native-PAGE, a band of approximately 160–200 kDa is also observed in the basal state, suggesting that higher-order multimers of NINJ1 consistent with decamers or higher-order multimers may also be present in the plasma membrane of unstimulated macrophages (*Figure 3A*). In response to pyroptosis (nigericin in LPS-primed BMDM), necrosis (pneumolysin), and apoptosis (venetoclax) induction in primary BMDM, the endogenous NINJ1 signal shifts to a high molecular weight aggregate, suggestive of a clustering process. Importantly, this shift is completely abrogated by glycine treatment (*Figure 3A*). Using this same native-PAGE approach, glycine treatment of primary hMDMs (*Figure 4A*) and mouse RAW-ASC cells (*Figure 1—figure supplement 1D*) undergoing pyroptosis prevented the shift of NINJ1 to a high molecular weight band. Finally, glycine inhibited NINJ1 oligomerization and pyroptotic lysis (as measured by LDH release) in LPS-primed and nigericin-treated human macrophages derived from induced pluripotent stem cells (iPSCs) derived macrophages (iPSDMs) without impairing secretion of IL-1β, confirming previous findings that glycine acts downstream of GSDMD pore formation (*Figure 4—figure supplement 2B–D*).

To corroborate these biochemical findings, we proceeded to evaluate native NINJ1 oligomerization within the plasma membrane by total internal reflection fluorescence (TIRF) microscopy in primary BMDM and hMDMs induced to undergo pyroptosis. TIRF microscopy allows for the preferential excitation of a thin (~150 nm) layer that includes the ventral plasma membrane of adherent cells, thereby eliminating background fluorescence from structures outside this focal plane (e.g. endomembrane compartments). In LPS-primed BMDMs and hMDMs, NINJ1 appears in discrete, small puncta that are diffusely distributed across the plasma membrane in both cell types (*Figure 3B*, *Figure 4B*). Upon induction of pyroptosis, the density of these puncta decreases, and their intensity increases (*Figure 3C–D*, *Figure 4C–D*), consistent with the clustering or aggregation of the low-order multimers into high-order oligomers as seen on the native PAGE (*Figure 3A*, *Figure 4A*). Co-treatment with glycine abrogates the formation of these clusters (*Figure 3B–D*, *Figure 4B–D*).

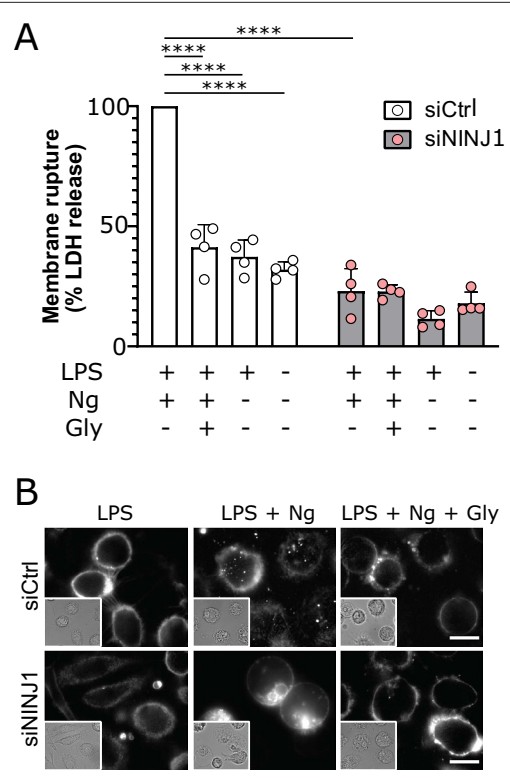

**Figure supplement 1.** Glycine dose-dependently inhibits rupture of pyroptotic human monocyte-derived macrophages (hMDMs).

**Figure supplement 1—source data 1.** Numerical values for experimental data plotted in *Figure 2—figure supplement 1*.

**Figure supplement 2.** Glycine does not confer cytoprotection in necroptotic human macrophages.

**Figure supplement 2—source data 1.** Numerical values for experimental data plotted in *Figure 2—figure supplement 2*.

**Figure supplement 3.** Glycine preserves membrane integrity but not cell viability in human macrophages undergoing pyroptosis.

**Figure supplement 3—source data 1.** Numerical values for experimental data plotted in *Figure 2—figure supplement 3*.

**Figure 2.** NINJ1 silencing in primary human monocyte-derived macrophages (hMDMs) functionally and morphologically phenocopies glycine cytoprotection. Primary hMDMs were treated with siRNA targeting *NINJ1* (siNINJ1) or non-targeted control (siCtrl). (**A**) Wildtype and *NINJ1*-silenced hMDMs were LPS-primed and induced to undergo pyroptosis (nigericin) with or without 50 mM glycine. LDH release in the supernatants was measured to assess cytotoxicity. Cytotoxicity was reduced in glycine-treated wildtype cells, as well as in *NINJ1*-silenced cells. Glycine treatment did not yield additional protection from pyroptotic cell death in *NINJ1*-silenced cells. Data are expressed as % of LDH in supernatant from siCtrl-treated cells stimulated to undergo pyroptosis from n=4 independent donors. Data points from each donor are shown along with their mean and SD. **** p<0.0001 by two-way ANOVA with Tukey's multiple comparison correction. (**B**) Nigericin- and glycine-treated LPS-primed wildtype hMDMs show similar ballooning morphology as nigericin-treated LPS-primed *NINJ1*-silenced hMDMs. Membrane ballooning is shown in live cells labeled with the plasma membrane dye FM1-43. The corresponding brightfield image is shown in the inset. Scale bar 20 μm.

The online version of this article includes the following source data and figure supplement(s) for figure 2:

**Source data 1.** Numerical values for experimental data plotted in *Figure 2*.

While the use of TIRF microscopy allowed us to localize our evaluation of NINJ1 clustering to the plasma membrane, it inadequately resolved the spatial clustering due to its diffraction limitation. To partially overcome this limitation, we next probed NINJ1 membrane clustering using stimulated emission depletion (STED), a super-resolution fluorescence microscopy approach, in primary BMDM. With STED, we improved upon the resolution from a full-width half-maximum by confocal microscopy of 271±98 nm to 63±8 nm (*Figure 3E*). In the basal state, NINJ1 was observed in small discrete puncta, which decreased in density (*Figure 3F*) and increased in intensity (*Figure 3G*) upon pyroptosis induction with nigericin. Co-treatment with glycine (5 mM) limited clustering (*Figure 3F–G*). Together, our biochemical and microscopy-based data are consistent with the notion that during pyroptosis, low-order NINJ1 oligomers cluster within the plasma membrane, a phenomenon suppressed by glycine.

## Glycine targets NINJ1 clustering to confer cytoprotection independently of upstream programmed lytic cell death signaling

Thus far, our data do not rule out the possibility that glycine is able to target proteins upstream of NINJ1 in pyroptosis. To test whether this is indeed the case, we evaluated caspase-1 activation, GSDMD processing, and IL-1β processing and secretion in iBMDM induced with nigericin with and without 5 mM glycine. Co-treatment

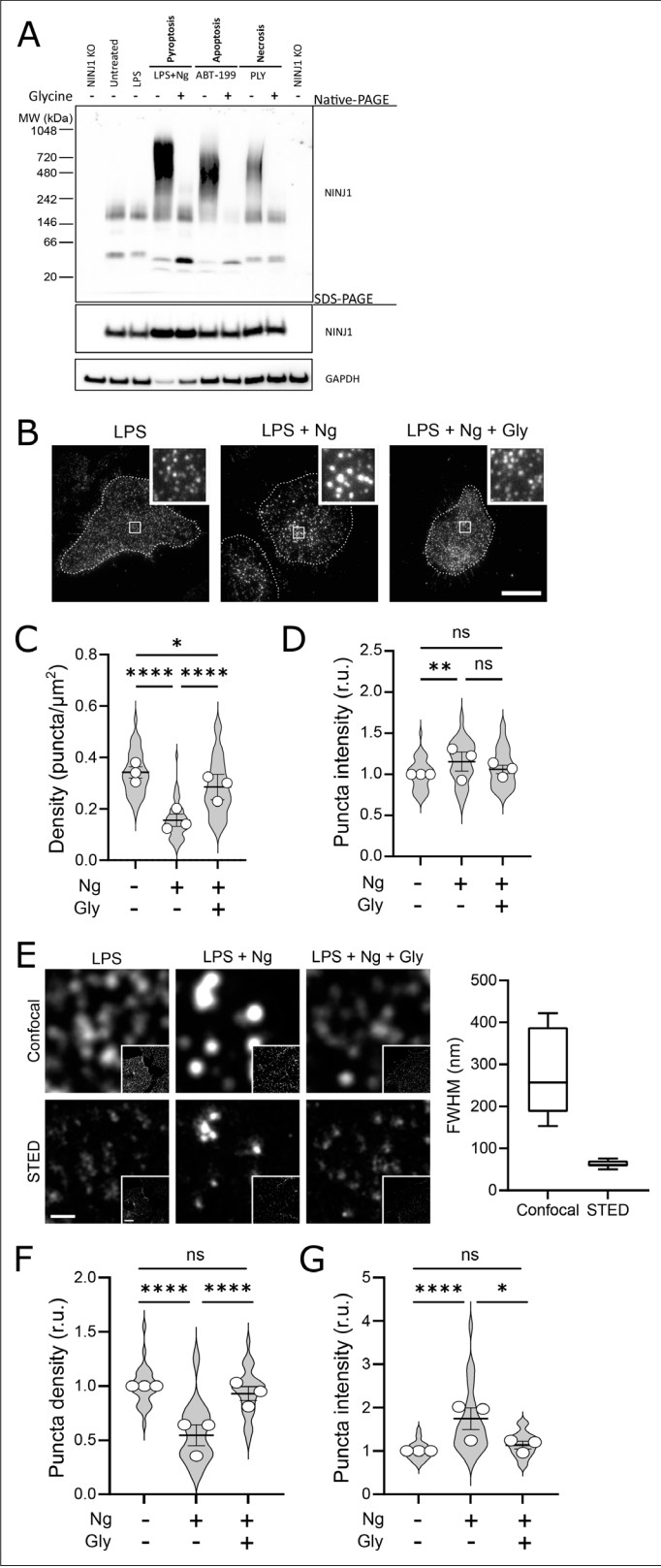

**Figure 3.** Glycine targets NINJ1 oligomerization to prevent membrane rupture in primary mouse macrophages. (**A**) Primary bone marrow-derived macrophage (BMDM) was induced to undergo pyroptosis (nigericin), apoptosis (venetoclax, ABT-199), or necrosis (pneumolysin) with or without glycine treatment. Native-PAGE analysis of endogenous NINJ1 demonstrates a shift to high molecular weight aggregate upon cell death stimulation, which

*Figure 3 continued on next page*

*Figure 3 continued*

is abrogated by glycine treatment. (**B**) Total internal reflection microscopy of NINJ1 in LPS-primed primary BMDM reveals that endogenous NINJ1 resides in discrete puncta within the plasma membrane. Cell membrane outline (dotted white line) was determined using fluorescently labeled cholera toxin subunit B (not shown) as a plasma membrane marker. Scale bar 20 μm. Inset shows magnified area demarcated by the white box. Anti-mouse NINJ1 antibody validation for immunofluorescence is provided in *Figure 3—figure supplement 1*. Quantification of the density (**C**) and mean fluorescence intensity (**D**) of NINJ1 puncta in LPS-primed primary macrophages at baseline or stimulated to undergo pyroptosis (nigericin 20 μM for 30 min) without or with glycine (5 mM). NINJ1 puncta become less dense and brighter upon pyroptosis induction, consistent with NINJ1 plasma membrane clustering. Glycine limits this redistribution. Violin plot of NINJ1 puncta density from three pooled independent experiments. Data points superimposed on the violin plots are the mean NINJ1 puncta densities for the three independent experiments (10–23 cells measured per replicate with >675 NINJ1 puncta identified per replicate). Bars represent mean ± SEM of the NINJ1 densities for the n=3 independent experiments. * p<0.05, ** p<0.01, and **** p<0.0001 by Kruskal–Wallis test with Dunn's multiple comparison correction. (**E**) Representative confocal (top) and stimulated emission depletion (STED; bottom) images of LPS-primed primary mouse bone marrow-derived macrophage (BMDM) at baseline or stimulated to undergo pyroptosis (nigericin 20 μM for 30 min) without or with glycine (5 mM). Full-width at half maximum of identified NINJ1 puncta shows a resolution of 271±98 nm and 63±8 nm (mean ± SD) for standard confocal and STED microscopy, respectively. Scale bar 0.5 μm; inset scale bar 20 μm. (**F–G**) Quantification of the normalized NINJ1 puncta density (**F**) and mean fluorescence intensity (**G**) from the STED images. The NINJ1 puncta become less dense with a corresponding increase in fluorescence intensity upon pyroptosis induction, consistent with NINJ1 plasma membrane clustering. Glycine limits these effects. Violin plot of the individual NINJ1 puncta from three pooled independent experiments (8–14 cells per replicate with >4964 NINJ1 puncta identified per replicate). Bars represent mean ± SEM of the NINJ1 densities for the n=3 independent experiments. * p<0.05 and **** p<0.0001 by Kruskal–Wallis test with Dunn's multiple comparison correction.

The online version of this article includes the following source data and figure supplement(s) for figure 3:

**Source data 1.** Numerical values for experimental data plotted in *Figure 3*.

**Figure supplement 1.** Rabbit anti-mouse NINJ1 monoclonal antibody validation.

with glycine had no effect on these upstream processes (*Figure 5A–B*), supporting the hypothesis that the target of glycine is downstream and independent of GSDMD activation and oligomerization. To confirm that glycine interferes with NINJ1-mediated cellular rupture independent of upstream signaling pathways, we generated a doxycycline-inducible NINJ1 overexpression system in mouse macrophages (*Figure 5—figure supplement 1*) based on the rationale that sufficient overexpression of NINJ1 can drive spontaneous lysis as seen in 293T systems (*Kayagaki et al., 2021*). Using this cell-based system, we demonstrate that glycine dose-dependently suppresses NINJ1-mediated cell rupture independent of activation of upstream programmed cell death pathway activation (*Figure 5C*). Expression of N-terminal GSDMD using a similar inducible system was able to trigger a glycine-inhibitable cellular rupture (*Figure 5D*). To further address glycine's mode of action, we evaluated glycine's effect on NINJ1 clustering biochemically and found that glycine suppresses NINJ1 clustering when NINJ1 is overexpressed (*Figure 5E*).

We next turned our attention to the purported amphipathic α-helix in the NINJ1 N-terminus (residues 40–69 of human NINJ1), which others have shown to be necessary for NINJ1-mediated plasma membrane rupture (*Kayagaki et al., 2021*). Circular dichroism (CD) of the α-helix at varying glycine concentrations between 0 and 20 mM did not affect the CD spectrum, indicating that at the trialed concentrations, glycine did not alter the stability of the helical secondary structure of this region of the NINJ1 N-terminus (*Figure 5—figure supplement 2*). We then functionally evaluated whether glycine interferes with the membrane lytic function of the helix using a reconstituted liposomal rupture assay. In this system, we demonstrate that while the α-helix was able to rupture liposomes, glycine did not interfere with this effect (*Figure 5F*). These data suggest that glycine can block full-length NINJ1 clustering and membrane rupture in cellulo but that additional unknown players may be affected within cells, or the mode of action is separate from the minimally sufficient lytic α-helix of the NINJ1 N-terminus as shown by intact liposome rupture.

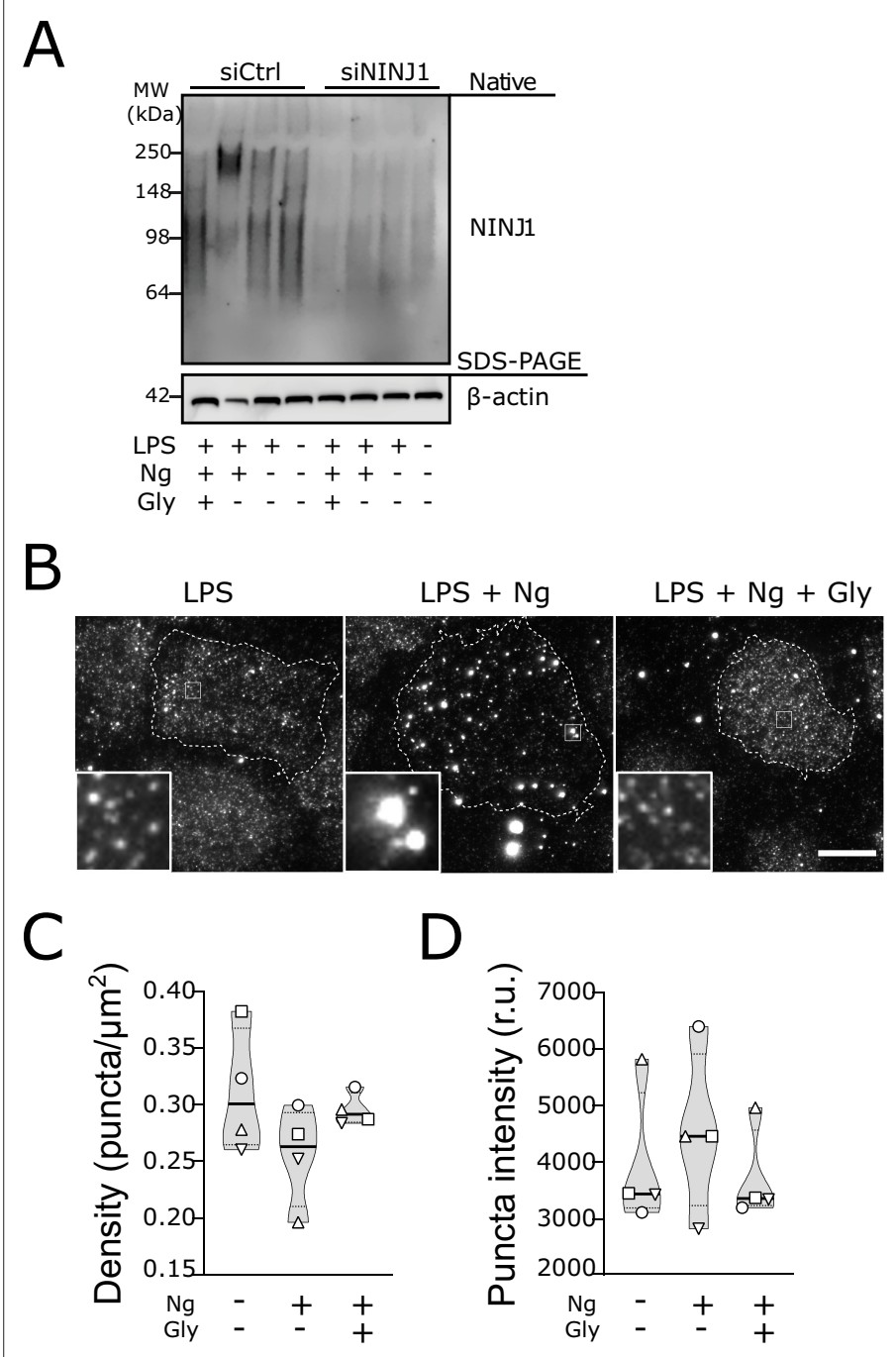

**Figure 4.** Glycine targets NINJ1 oligomerization to prevent membrane rupture in primary human macrophages. (**A**) Native-PAGE analysis of endogenous NINJ1 from human monocyte-derived macrophages (hMDMs) stimulated to undergo pyroptosis displays a shift to higher molecular weight, which is abrogated by glycine treatment. NINJ1 levels are almost absent in siNINJ1-treated cells (quantification in *Figure 4—figure supplement 2A*), and no shift is seen after LPS and nigericin treatment. (**B**) Total internal reflection fluorescence (TIRF) microscopy of LPS-primed primary hMDMs shows endogenous NINJ1 in discrete plasma membrane puncta. Cell membrane outline (dotted white line) was determined using fluorescently labeled cholera toxin subunit B (not shown) as a plasma membrane marker. Scale bar 20 µm. Anti-human NINJ1 antibody validation for immunofluorescence is provided in *Figure 4—figure supplement 1*. Quantification of the NINJ1 puncta density (**C**) and fluorescence intensity (**D**) in LPS-primed hMDMs at baseline or stimulated to undergo pyroptosis (nigericin 20.7 µM for 2 hr) without or with glycine (50 mM). Violin plot of NINJ1 puncta density and intensity quantified from images of cells from four independent

*Figure 4 continued on next page*

*Figure 4 continued*

donors. Mean of NINJ1 puncta densities from three cells per donor per condition shown as superimposed datapoints (different symbols are used for individual donors) along with median and quartiles for each condition.

The online version of this article includes the following source data and figure supplement(s) for figure 4:

**Source data 1.** Numerical values for experimental data plotted in *Figure 4*.

**Figure supplement 1.** Mouse anti-human NINJ1 monoclonal antibody validation.

**Figure supplement 2.** Glycine prevents NINJ1 aggregation and pyroptotic lysis in induced pluripotent stem-cell (iPSC) derived human macrophages (iPSDMs) without affecting IL-1β secretion.

**Figure supplement 2—source data 1.** Numerical values for experimental data plotted in *Figure 4—figure supplement 2*.

## Discussion

The protective effect of glycine treatment is a pervasive feature of many cell death pathways, yet the mechanism of action was previously unknown. Here, we position the plasma membrane clustering of the pan-death protein NINJ1 as an elusive glycine target that mediates cytoprotection.

In our studies, we focused on fundamental cell death pathways in both mouse and human macrophages and show that glycine prevents NINJ1 clustering, a process required for plasma membrane rupture. Necroptosis represents another programmed cell death pathway, important to inflammatory responses, pathogen detection, and tissue repair (*Bertheloot et al., 2021*). Unlike pyroptosis, necrosis, and post-apoptosis lysis, NINJ1 does not appear to be needed for necroptosis-induced plasma membrane rupture (*Kayagaki et al., 2021*), potentially due to the membrane-disrupting capacity of the necroptosis MLKL pore (*Flores-Romero et al., 2020*). This is particularly notable as glycine does not prevent TNF-induced necroptosis (*Figure 2—figure supplement 2* and *Chen et al., 2014*), further buttressing the association between NINJ1 and glycine.

Our data demonstrate that glycine treatment and genetic ablation or silencing of NINJ1 yield similar functional and morphological outcomes in both human and mouse macrophages subjected to lytic cell death programs. Mechanistically, our data suggest that glycine cytoprotection results from interference with NINJ1 clustering within the plasma membrane. This finding is further supported by the observations that (1) glycine treatment of NINJ1 knockout cells confers no additional protection against cellular rupture; (2) glycine cytoprotection is independent of upstream programmed lytic cell death signaling; and (3) amino acids structurally similar to glycine inhibit NINJ1 clustering and membrane rupture with the same potency hierarchy as previously reported for glycine cytoprotection in pyroptosis (*Figure 1—figure supplement 1B–C* and *Loomis et al., 2019*).

The mechanism by which glycine interferes with NINJ1 clustering can be either direct or indirect. Our in vitro CD studies and liposomal rupture system, respectively, indicate that glycine does not interfere with the secondary structure or lytic function of the NINJ1 α-helix. It remains plausible that glycine still interacts directly with this domain or other components of the NINJ1 N-terminus in cells to interfere with NINJ1 clustering, consistent with our presented data. Alternatively, glycine may act on an unidentified intermediate that modulates NINJ1 clustering within the plasma membrane. Our data in pyroptotic cells suggest that such an intermediate would reside between active GSDMD and NINJ1. In an indirect model, any of a plasma membrane-associated protein, membrane lipid, or cellular metabolic pathway could be the target. In addition, it remains unclear whether glycine's cytoprotective activity occurs on the intracellular or extracellular side of the plasma membrane. To the best of our knowledge, there is no direct evidence to suggest whether glycine's cytoprotective effect is specifically mediated from the intracellular or extracellular side. Both extracellular (*Weinberg et al., 2016*) and intracellular (*Rühl and Broz, 2022*) sites of action have been proposed. Our in vitro data suggest that glycine is not acting directly on the N-terminal α-helix, which is thought to be extracellular; however, we do not otherwise delineate the topology of glycine's mechanism of action.

Defining the site of action would provide insight into the molecular mechanism by which glycine inhibits NINJ1 clustering in the plasma membrane and ultimately help determine whether glycine directly or indirectly engages NINJ1. Indeed, how NINJ1 is activated by lytic cell death pathways, its clustering trigger, and method of plasma membrane disruption also remain important and outstanding questions in the field. Due to the ubiquity of lytic cell death pathways in human health and disease and the potency of glycine cytoprotection, answers to these questions will greatly advance the

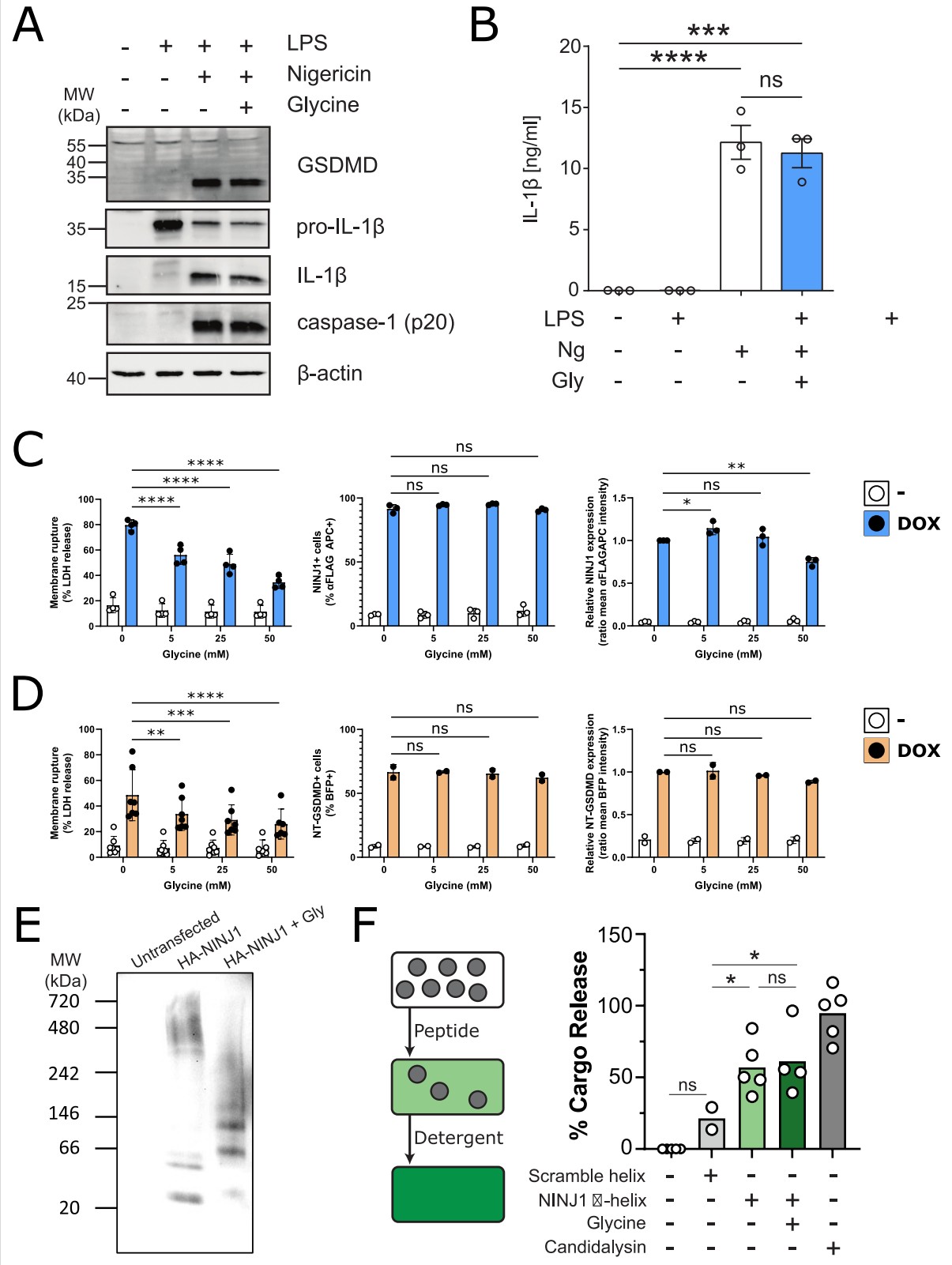

**Figure 5.** Glycine inhibits NINJ1 clustering but not upstream pyroptosis signaling to confer cytoprotection. (**A–B**) Wildtype immortalized bone marrow-derived macrophage (iBMDM) was primed with 1 µg/mL LPS for 4 hr or left unprimed before stimulation of primed cells with 20 µM nigericin for 2 hr in the presence or absence of 5 mM glycine. (**A**) Processing of GSDMD, caspase-1, and IL-1β was assessed by western blot. (**B**) Release of IL-1β into the cell culture supernatant was quantified by ELISA. ELISA results show mean ± SEM of three independent experiments. Western blots are representative

*Figure 5 continued on next page*

*Figure 5 continued*

of three independent experiments. ***p<0.001 and ****p<0.0001 by one-way ANOVA with Tukey's multiple comparison correction. (**C–D**) Glycine dose-dependently inhibits plasma membrane rupture in mouse macrophages overexpressing NINJ1 or the N-terminal fragment of GSDMD. iBMDMs with doxycycline-inducible expression of FLAG-tagged NINJ1 (**C**) or the N-terminal fragment of GSDMD fused to BFP(blue fluorescent protein) (**D**) were incubated with 2 µg/mL doxycycline and increasing concentrations of glycine for 12 hr or 8 hr, respectively, before analyses of cell rupture (LDH release) and protein expression. Left: LDH release in supernatants relative to full lysis controls. Middle: frequencies of NINJ1-FLAG-positive or GSDMD-NT-BFP-positive cells. Right: surface expression of NINJ1-FLAG or GSDMD-NT-BFP by mean fluorescence intensity. * p<0.05, **p<0.01, ***p<0.001, and ****p<0.0001 by two-sided ANOVA with Tukey's multiple comparison correction. (**E**) HeLa cells were transiently transfected with HA-tagged mouse NINJ1 in the presence or absence of glycine (5 mM). Native-PAGE analysis of ectopically expressed NINJ1 demonstrates a shift to high molecular weight aggregate, which is abrogated by glycine treatment. (**F**) Large unilamellar vesicles (LUVs; gray circles) were made containing 25 mM carboxyfluorescein (CF) at which the CF fluorescence self-quenches. NINJ1 α-helix peptide (corresponding to amino acids 40–69 of human NINJ1) is added to the LUV suspension. Ruptured LUV release CF, which no longer self-quenches. The resulting increase in fluorescence is monitored using a spectrofluorometer. Detergent is added to rupture all remaining liposomes to capture the maximum attainable fluorescence. LUV rupture by N-terminus NINJ1 α-helix without and with glycine (50 mM), scrambled NINJ1 α-helix peptide, or the cytolytic yeast peptide candidalysin compared to vehicle. Glycine does not prevent LUV rupture by the N-terminal NINJ1 α-helix. * p<0.05 by ANOVA with Tukey's multiple comparison correction.

The online version of this article includes the following source data and figure supplement(s) for figure 5:

**Source data 1.** Numerical values for experimental data plotted in *Figure 5*.

**Figure supplement 1.** Characterization of the inducible immortalized bone marrow-derived macrophage (iBMDM)-doxy-moNINJ1 system.

**Figure supplement 1—source data 1.** Numerical values for experimental data plotted in *Figure 5—figure supplement 1*.

**Figure supplement 2.** Circular dichroism (CD) spectroscopy of human NINJ1 peptide reveals a stable α-helical secondary structure insensitive to glycine.

**Figure supplement 2—source data 1.** Numerical values for experimental data plotted in *Figure 5—figure supplement 2*.

development of therapeutics against pathologic conditions associated with aberrant lytic cell death pathways.

# Materials and methods
## Cells

Primary BMDMs were harvested from the femurs and tibia of wildtype mixed-sex cohorts of C57Bl/6 mice. The ends of cleaned bones were cut and centrifuged to collect bone marrow into sterile PBS. The cell suspension was washed with PBS and plated in DMEM with 10 ng/mL M-CSF (315-02; Peprotech Inc, Cranbury, NJ, USA). Following 5 days of culture, the BMDM were detached from the dishes with TBS with 5 mM EDTA, resuspended in fresh DMEM, and plated. ASC-expressing RAW264.7 mouse macrophages (RAW-ASC; raw-asc, InvivoGen, San Diego, CA, USA) are engineered to stably express murine ASC, which is absent in the parental RAW 264.7 macrophage cell line (*Pelegrin et al., 2008*) and therefore are able to undergo pyroptosis. iBMDM expressing Cas9 was obtained from Dr Jonathan Kagan (Boston Children's Hospital, Boston) and described previously (*Evavold et al., 2018*). Cell lines were authenticated and negative for mycoplasma contamination.

## Isolation and differentiation of human primary macrophages

Buffy coats and pooled serum from healthy donors were provided by the blood bank at St. Olavs Hospital (Trondheim, Norway), after informed consent and with approval by the Regional Committee for Medical and Health Research Ethics (No. 2009/2245). Peripheral blood mononuclear cells were isolated by density gradient using Lymphoprep (Axis-Shield Diagnostics, StemCell Technologies, Cologne, Germany) according to the manufacturer's instructions. Monocytes were selected by α-CD14 bead isolation (Miltenyi Biotech, Lund, Sweden) and seeded in RPMI 1640 supplemented with 10% serum and 10 recombinant macrophage colony-stimulating factor (M-CSF; R&D Systems, Minneapolis, MN, USA) for 3 days. At day 3, the medium was replaced with RPMI 1640 without M-CSF, and siRNA treatment was initiated.

## iPSDMs culture and production

iPSCs were obtained from European Bank for induced pluripotent Stem Cells (EBiSC, https://ebisc.org/about/bank), distributed by the European Cell Culture Collection of Public Health England (Department of Health, United Kingdom). iPSCs were maintained in Vitronectin XF (StemCell Technologies) coated plates with mTeSR Plus medium (StemCell Technologies). Cells were passaged with ReLeSR (StemCell Technologies) and plated in media containing 10 μM Rho-kinase inhibitor Y-27632 (StemCell Technologies). Embryonic body (EB) formation and myeloid differentiation are based on and adapted from a previously reported protocol (*van Wilgenburg et al., 2013*). Briefly, iPSCs were washed with PBS before Versene (Gibco-Thermo Fisher, Waltham, MA, USA) was added and cells incubated at 37°C for 5 min to generate a single cell suspension of iPSCs. Cells were counted, washed with PBS, and resuspended to a final concentration of 10,000 cells/50 μL in EB media: mTeSR Plus medium (StemCell Technologies), 50 ng/mL BMP-4 (R&D Systems, 315 BP-010), 20 ng/mL SCF (R&D Systems, 255-SC-050), and 50 ng/mL VEGF (R&D Systems, 293-VE-050). Cell suspension, supplemented with 10 μM Y-27632, was added into a 384-well Spheroid Microplate (Corning. #3830), 50 μL/well. The microplate was centrifuged at 1000 RPM for 3 min, and incubated at 37°C, 5% $CO_2$ for 4 days. EBs were fed at day 2 by adding 50 μL of fresh EB medium. After 4 days, EBs were harvested and seeded into a gelatin coated T175 flask in X-VIVO-15 (Lonza) media, supplemented with 100 ng/mL M-CSF (Preprotech, 300–25), 25 ng/mL IL-3 (Preprotech, 200–03), 2 mM glutamax (Gibco-Thermo Fisher), and 0.055 mM β-mercaptoethanol (Gibco-Thermo Fisher). Media was changed every week, and EB-culture was renewed after 4 months. Monocytes produced were routinely checked with regards to phenotype using a defined flow cytometry panel of expected surface markers. Monocytes were harvested once a week and differentiated into macrophages in RPMI 1640 with 10% fetal calf serum and 100 ng/mL M-CSF for 5–7 days.

## NINJ1 knock-out cell line generation and siRNA silencing of NINJ1

RAW-ASC and iBMDM cells were transfected with custom CRISPR gRNA plasmid DNA (U6-gRNA:CMV-Cas-9–2 A-tGFP; Sigma-Aldrich, Oakville, Canada) using FuGENE HD transfection reagent (Promega, Madison, WI, USA). The *Ninj1* target region sequence was GCCAACAAGAAGAGCGCTG. 24 hr later, the cells were FACS sorted for GFP into 96 well plates. Individual colonies were expanded and tested for NINJ1 expression by western blot. NINJ1 knockdown in hMDMs was performed by siRNA using Lipofectamine RNAiMAX (Invitrogen, Waltham, MA, USA) according to the manufacturer's instructions. Cells were transfected with 20 nM pooled siRNA against *NINJ1* (Thermo Scientific, HSS107188, HSS107190, HSS181529) or non-silencing control siRNA (Qiagen, Hilden, Germany, 1027310). hMDMs were siRNA-treated two times (day 3 and day 5) before medium was changed to RPMI with 1% serum (day 6), and cells were allowed to rest overnight prior to stimulation. Knockdown of NINJ1 was confirmed by western blotting.

## Generation of N-terminal HA-NINJ1 construct and cellular expression

PCR reactions were performed using ConeAmp HiFi master mix (Takara). Primers for mouse *Ninj1* (insert) PCR reaction were: forward primer 5'gat tac gct gag tcg ggc act gag ga3' and reverse primer 5'ctc ggt acc cta ctg ccg ggg cgc ca3'. Template DNA for the PCR reaction was mouse *Ninj1* cDNA clone from Origene (Cat. No. MR225037). The template vector pcDNA3.1-HA was obtained from Addgene. Primers for pcDNA3.1-HA PCR (vector) reaction were: forward primer 5'cgg cag tag ggt acc gag ctc gga3' and reverse primer 5'gcc cga ctc agc gta atc tgg aac atc3'. PCR reaction products were cleaned up using the NucleoSpin PCR Clean-up kit (Macherey-Nagel). The PCR products contained 18 bp overlapping regions between the pcDNA3.1-HA tag vector and the mouse NINJ1 insert. This allows for fusion of the insert and vector using the In-Fusion Snap Assembly EcoDry Cloning Kit (Takara, 638954) and transformation into Stellar competent bacterial cells (Takara). DNA sequencing with CMV-F primer confirmed *Ninj1* insertion into the pcDNA3.1-HA vector and that NINJ1 was in-frame with the N-terminal HA-tag. HeLa cells (ATCC #CCL-2) were transfected with pcDNA 3.1HA-NINJ1 using Fugene HD transfection reagent (Promega) according to the manufacturer's protocol.

## Inducible iBMDM-doxy-moNINJ1 system

iBMDMs with doxycycline-inducible expression of the N-terminal fragment of GSDMD fused to BFP are published (*Evavold et al., 2021*). We used the same stock of iBMDMs from Cas9-expressing mice in which the TET3G transactivator protein had introduced previously (*Evavold et al., 2021*) to generate cell lines with dox-inducible expression of NINJ1 carrying an N-terminal FLAG-tag. cDNA encoding full-length murine NINJ1 was ordered as a gBlock from Integrated DNA Techologies. This sequence was amplified by PCR to append an N-terminal FLAG-tag and cloned into pRETROX-TRE3G (TakaraBio) using BamHI and NotI restriction sites and into pMSCV IRES EGFP using NotI and SalI restriction sites. To introduce dox-inducible FLAG-NINJ1, $2.5 \times 10^6$ Platinum-GP retrovirus packaging cells were seeded in a TC-treated 10 cm dish. On the next day, cells were transfected with 3 µg pCMV-VSV-G and 9 µg pRETROX-TRE3G encoding FLAG-NINJ1 using Lipofectamine 2000. After 18 hr, media was exchanged to 6 mL of fresh complete DMEM. The same day, $0.5 \times 10^6$ Cas9- and TET3G-expressing iBMDMs were seeded per well of 6-well plate. The next day, retroviral supernatant was harvested and filtered through 0.45 µm syringe filter, supplemented with polybrene (1:2000; EMD Millipore), and spun onto cells (1250 g, 30°C, 1 hr). 6 mL of fresh cDMEM were added to retrovirus packaging cells, and spinfection was repeated the next day. 3 days after the second transduction, transduced cells were selected by growth in complete DMEM including 1.5 mg/mL G-418 and 10 µg/mL puromycin. To ensure maximal transgene expression, which correlates with the ability of FLAG-NINJ1 to induce cell death and lysis, FLAG-NINJ1 was also stably overexpressed in these cells using retroviral transduction with an MSCV(murine stem cell virus) retrovirus. Retrovirus production and spinfections were performed as described above, only using pMSCV IRES EGFP-FLAG-NINJ1 instead of pRETROX-TRE3G construct. Cells were then sorted on a BD Melody cell sorter for maximal EGFP expression (top 5–10%). Clonal cell lines were derived by limited serial dilution.

$1 \times 10^5$ cells per well in a black 96-well plate in DMEM (Gibco)/10% FBS(fetal bovine serum) (R&D, S11550) were induced with 2 µg/mL doxycycline (Sigma-Aldrich) in the presence or absence of increasing concentrations of glycine (Sigma-Aldrich 67126) in a total volume of 200 µL for 8 hr (iBMDM GSDMD-NT) or 12 hr (iBMDM FLAG-NINJ1) before supernatants were harvested for LDH-assay (Invitrogen) and cells detached (PBS/4 mM EDTA) for flow cytometry (Becton Dickinson Fortessa). iBMDM GSDMD-NT express BFP and were analyzed directly; iBMDM NINJ1 were stained with anti-FLAG APC antibody (1:100, Abcam Ab 72569; RRID: AB_1310127) in PBS/2% FBS, 2.5 mM EDTA on ice and washed once before analysis.

## Pyroptosis, apoptosis, necrosis, and necroptosis induction

Pyroptosis was activated in BMDM primed with (0.5 µg/mL) LPS from *Escherichia coli* serotype 055:B5, which was first reconstituted at a stock concentration of 1 mg/mL. In primary BMDM, after 4.5 hr of LPS priming, pyroptosis was induced with 20 µM nigericin (Sigma N7143; stock 10 mM in ethanol) for 30 min. iBMDM were primed with LPS for 3 hr, followed by pyroptosis induction with 10 µM nigericin for 2 hr. Apoptosis was induced in non-primed macrophages using the Bcl-2 inhibitor venetoclax (ABT-199; Tocris, Bristol, United Kingdom; 6960) at 25 µM for 16 hr. Necrosis was induced by treating non-primed macrophages with pneumolysin (0.5 µg/mL) for 15 min in iBMDM or 45 min in primary BMDM. Pneumolysin was obtained from Dr John Brumell (Hospital for Sick Children, Toronto). In hMDMs, pyroptosis was induced by priming with 1 µg/mL LPS from *E. coli* serotype 0111:B4 for 2 hr, followed by stimulation with 20.7 µM nigericin (Invivogen tlrl-nig; stock 5 mg/mL in ethanol) for 2 hr and 15 min. Necroptosis was induced by treating cells with pan-caspase inhibitor zVAD (25 uM, Invivogen, tlrl-vad) and SMAC mimetic BV6 (10 uM, Invivogen, inh-bv6) for 30 min, before stimulation with recombinant human TNF-α (10 ng/mL, Peprotech, 300–01 A) for 4 hr. iPSDMs were stimulated with 200 ng/mL LPS for 3 hr and 20.7 µM nigericin for 1–1.5 hr.

Where indicated, mouse cells were treated with 5 mM glycine (Sigma, G7126) at the time of cell death induction. Human cells were pretreated with glycine (Millipore, Burlington, MA, USA; 104201) at a concentration of 50 mM for at least 10 min before stimuli was added.

## LDH assay

BMDM were seeded at 200,000 cells per well in 12-well plates, treated as indicated, and cytotoxicity assayed by LDH release assay the following day. At the end of the incubations, cell culture supernatants were collected, cleared of debris by centrifugation for 5 min at 500× *g*. The cells were washed

once with PBS then lysed in lysis buffer provided in the LDH assay kit (Invitrogen C20300). Supernatants and lysates were assayed for LDH using an LDH colorimetric assay kit as per the manufacturer's instructions (Invitrogen C20300). hMDMs were seeded at 500,000 cells per well in 12-well plates and differentiated as described above. iPSDMs were seeded at 180,000–450,000 cells per well in 12-well plates and differentiated as described above. After cells were stimulated as described in figure legends, supernatants were harvested, and LDH release was assayed (Invitrogen C20301). Briefly, equal amounts of supernatant and reaction mixture were mixed and incubated for 30 min, followed by absorbance reads at 490 and 680 nm. To calculate the LDH release, the 680 nm background was subtracted from the 490 nm absorbance reads before further processing.

## Western blot cleavage GSDMD, caspase-1, and IL-1β

iBMDMs were seeded at $10^6$ cells per well in 12-well plates overnight, primed with 1 µg/mL *E. coli* LPS (Enzo; Serotype O111:B4) for 4 hr, or left unprimed before stimulation with 20 µM of nigericin (Invivogen) in 500 µL Opti-MEM (Gibco) with no serum. After 2 hr, 50 µL of supernatant was collected for analysis by ELISA. 125 µL of concentrated 5× SDS loading buffer with reducing agent TCEP(tris(2-carboxyethyl)phosphine) was added to each well to stop the reaction. This treatment ensured the capture of proteins in both the supernatant and the cell lysates for western blot analysis. Proteins were separated by SDS-PAGE and detected by western blot using primary antibodies against GSDMD (Abcam ab209845; RRID: #AB_2783550), Caspase-1 (Adipogen AG-20B-0042-C100; RRID: #AB_2755041), pro-IL-1β (Genetex GTX74034; RRID: #AB_378141), and actin (Sigma A5441; RRID: #AB_476744).

## Mitochondrial membrane potential and cellular ATP measurements

hMDMs were differentiated at 30,000 cells per well in glass-bottom 96-well plates (Cellvis, P96-1.5H-N). Parallel plates were prepared for live-cell imaging and LDH + CellTiter Glo analysis. hMDMs were either left untreated or pre-treated with glycine (50 mM) and primed with 1 µg/mL LPS (*E. coli* serotype 0111:B4) in RPMI complemented with 1% fetal calf serum for 3 hr before stimulation with 20.7 µM of nigericin (Invivogen). Tetramethylrhodamine ethyl ester perchlorate (TMRE, 200 or 10 nM, Sigma-Aldrich) was added to cells before the plates were placed in a heated incubator (37°C, 5% $CO_2$), and images were taken every 30 min over 18 or 22 hr (two independent experiments). In the second experiment, at the end of LPS priming, cells were labeled with CellTracker Deep Red (0.5 µM Invitrogen) for 15 min and washed before addition of fresh medium with TMRE (10 nM) with or without LPS, nigericin, and glycine. In parallel plates treated identically, supernatants were harvested and assayed for LDH, and cellular ATP was measured using CellTiterGlo (Promega) at 0, 2, 6, and 18/22 hr.

## Fluorescence microscopy

Mouse macrophages were cultured on 18 mm glass coverslips in 12-well plates at 150,000 cells per well. Cells were treated as indicated in the figure legends, washed with PBS, fixed in 4% paraformaldehyde (PFA) in PBS at room temperature for 15 min, and permeabilized with 0.1% Tween-20. Cells were then blocked in PBS supplemented with 10% donkey serum and 0.1% Tween-20 for 1 hr at room temperature prior to overnight incubation with primary antibody at 4°C. Rabbit monoclonal anti-mouse NINJ1 was used at 10 µg/mL. Next, cells were washed three times with PBS supplemented with 10% donkey serum prior to the addition of secondary antibody at room temperature for 1 hr. In validation studies of the rabbit anti-mouse NINJ1 monoclonal antibody, cells were stained with and without the primary antibody prior to application of the secondary antibody. Nuclei were labeled using DAPI containing mounting medium (ProLong Diamond Antifade Mountant, Invitrogen P36961). For total internal reflection microscopy, cells were co-stained with recombinant Cholera toxin subunit B conjugated to Fluor488 (Invitrogen C34775) at 1 µg/mL for less than 1 min prior to fixing and staining for NINJ1 as described above. The Cholera toxin subunit B served as a membrane marker to optimize plasma membrane visualization. TIRF imaging of mouse macrophages was conducted on a Zeiss Axio Observer Z1 microscope with a 100× objective and Andor iXon3 885 detector. For STED microscopy, primary mouse macrophages were cultured on 1.5 H 18 mm glass coverslips (Marienfeld) in 12-well plates at 150,000 cells per well. After treatments as indicated in the figure legend, cells were washed with PBS, then fixed in 4% PFA in PBS for 10–15 min at room temperature. Following fixation, cells were washed, then permeabilized in 0.1% Tween-20. Cells were blocked in in PBS supplemented

with 10% donkey serum and 0.1% Tween-20 for 1 hr at room temperature, followed by primary antibody incubation in blocking solution overnight at 4°C. Rabbit monoclonal anti-mouse NINJ1 was used at 10 µg/mL. The next day, cells were incubated with Cy3-conjugated secondary antibody (diluted 1:100) in 1% serum in PBST for 1 hr at room temperature. After final washes, coverslips were mounted onto glass slides using ProLong Diamond Antifade Mountant (Invitrogen, P36965). STED microscopy was performed using a Leica TCS SP8 STED 3× microscope using HyD detectors and a 100×/1.4 oil objective. Samples labeled with Cy3-conjugated secondary antibody were excited using a white light laser at 554 nm. Emissions were time-gated 1.30–6.00 ns. A 1.5 W depletion laser at 660 nm was used at 70% maximal power. Images were acquired and deconvolved using Leica LAS X software with Lightning Deconvolution with an *xy* pixel size of 12.62 nm. For all fluorescence microscopy in mouse macrophages, rabbit monoclonal anti-mouse NINJ1 was obtained from Drs Nobuhiko Kayagaki and Vishva Dixit (Genentech Inc, San Francisco, CA, USA). Where indicated, images of live cell morphology were captured using the epifluorescence mode and with FM 4–64 (Invitrogen, T3166) added right before imaging.

Human primary macrophages were differentiated at 30,000 cells per well in glass-bottom 96-well plates (Cellvis, P96-1.5H-N). After stimuli, the cells were washed with PBS and thereafter stained with Cholera toxin subunit B Alexa Fluor 488 conjugate at 1 µg/mL in PBS for less than 1 min. Cells were then fixed in 4% PFA in PBS at room temperature for 15 min, followed by washing three times with PBS. We permeabilized the cells with PBS containing 0.01% saponin for 10 min at room temperature and thereafter blocked with PBS/0.01% saponin/20% pooled human serum for 3 hr. Cells were either left unstained or incubated with anti-human NINJ1 antibody (mouse IgG2b, R&D Systems, MAB5105, RRID: AB_11128852) or a control Ab (mouse IgG2a, Serotec, MCA929F; RRID: AB_322271) at 10 µg/mL in PBS/0,01% saponin/1% pooled human serum at 4°C for 41 hr and then washed three times with PBS/0.01% saponin/1% pooled human serum. Following this, we incubated the cells with far-red fluorescent rabbit anti-mouse antibody (Invitrogen, A27029, RRID: AB_2536092) at 2 µg/mL in PBS/0.01% saponin/1% pooled human serum at room temperature for 1 hr. Cells were washed three times with PBS/0.01% saponin/1% pooled human serum and thereafter three times with PBS before imaging. TIRF imaging of human macrophages was done on a Zeiss TIRF 3 microscope with a 63× objective and a Hamamatsu ORCA Fusion sCMOS detector. Images of morphology in live cells were captured using the epifluorescence mode and with FM 1–43 (Invitrogen, T3163) added at 5 µg/mL right before imaging.

For TMRE experiments, human macrophages were imaged using a Zeiss TIRF III (Carl Zeiss GmbH, Jena) equipped with an alpha Plan-Apochromat 100×/1.46 oil-immersion objective and a Hamamatsu ORCA-Fusion (C14440-20UP) CMOS camera. A Plan-Apochromat 10×/0.45 air objective was used for the time series experiments. TMRE was excited by a Colibri.2 470 nm LED, and the emission was detected between 510–542 nm. CellTracker Deep Red was excited by a 638 nm diode laser, and the emission was detected between 665 and 711 nm. Image processing was done in FIJI (*Schindelin et al., 2012*). Automatic counting of TMRE-positive cells was done in CellProfiler v.4.2.4 (*Stirling et al., 2021*) by using the IdentifyPrimaryObjects module with either Maximum Cross-Entropy adaptive thresholding or Otsu thresholding, and settings were optimized for each dataset to enhance accuracy.

## Native and SDS-PAGE of NINJ1

Mouse macrophages were lysed with native-PAGE lysis buffer (150 mM NaCl, 1% Digitonin, 50 mM Tris pH7.5, and 1× Complete Protease Inhibitor). Following centrifugation at 20,800× *g* for 30 min, lysates were mixed with 4× NativePAGE sample buffer and Coomassie G-250 (ThermoFisher) and resolved using NativePAGE 3–12% gels. For SDS-PAGE, cells were washed with 1× PBS and lysed in RIPA lysis buffer containing protease inhibitors (Protease inhibitor tablet, Pierce A32955). Proteins were resolved using NuPAGE 4–12% Bis-Tris gels (Invitrogen), transferred to PVDF(polyvinylidene difluoride) membranes, and immunoblotted. For HA-NINJ1 overexpression experiments, rabbit monoclonal anti-HA (Cell Signaling Technology, Inc, Danvers, MA, USA; #3724; RRID: AB_1549585) was used. GAPDH (Santa Cruz Biotechnology, Dallas, TX, USA; sc-25778; RRID: AB_10167668 1:1000) was used for a loading control.

hMDMs and iPSDMs were lysed with 1× NativePAGE Sample Buffer containing 1% Digitonin. After centrifugation at 21,000× g for 30 min, protein concentration of lysates was measured using

Pierce BCA Protein Assay Kit (ThermoFisher) to ensure equal loading of proteins. For Native-PAGE on hMDMs, lysates were mixed with Novex Tris-Glycine Native Sample Buffer (Invitrogen) and resolved using Novex 4–12% Tris-Glycine gels (Invitrogen). For Native-PAGE on human iPSDMs, lysates were mixed with Coomassie G-250 (ThermoFisher) and resolved using NativePAGE 3–12% Bis-Tris gels (Invitrogen). For SDS-PAGE, lysates were mixed with DTT(dithiothreitol) and NuPAGE LDS sample buffer (Invitrogen), denatured for 5 min at 95 °C, and resolved using NuPAGE 4–12% Bis-Tris gels (Invitrogen). Proteins were transferred to PVDF membranes and immunoblotted. NINJ1 (R&D Systems, MAB5105, RRID: AB_11128852) was used to determine NINJ1 expression, while β-actin (Cell Signaling Technology, Danvers, MA, USA; Cat# 8457, RRID: AB_10950489) was used as loading control.

## Cytokine measurements

Supernatants from human iPSDMs and mouse iBMDM were assessed for IL-1β (R&D Systems, DY201 or Invitrogen 88–7013, respectively) by ELISA according to the protocol of the manufacturer.

## Large unilamellar vesicle rupture assay

To generate large unilamellar vesicles (LUVs), 1,2-dioleoyl-sn-glycero-3-phosphocholine (DOPC, Avanti Polar Lipids) and 1,2-dioleoyl-sn-glycero-3-phospho-l-serine (sodium salt) (DOPS, Avanti Polar Lipids) were first prepared at 80% DOPC and 20% DOPS, freeze dried, and hydrated with a solution containing 25 mM 6-carboxyfluorescein. The suspension was bath sonicated, freeze-thawed, and extruded (Avanti Mini Extruder) using a Nucleopore 0.1 µm membrane (Whatman) to yield LUVs, which were washed three times in PBS to remove excess 6-carboxyfluorescein. At this concentration, the fluorescence of 6-carboxyflurescein self-quenches. With liposomal rupture, the 6-carboxyfluorescein is released into the bathing solution, thereby releasing the self-quenching with fluorescence monitored in a spectrofluorometer. The liposomal rupture assay was setup by mixing equal amounts of the prepared LUVs with 0.4 mg/mL human NINJ1 α-helix region peptide (HYASKKSAAESMLDIA LLMANASQLKAVVE; GenScript Biotech Corporation) with either vehicle or glycine (50 mM) in PBS buffer. Where indicated, a sequence-scrambled analog peptide (IAAAAMKMYLANSLEHAKSLKVVL ASQDS; GenScript Biotech Corporation) was used as a negative control for the human NINJ1 α-helix region peptide. Following 45 min, all liposomes were then ruptured by adding Triton X-100 and the maximum fluorescence (cargo release) recorded. The results were converted to percentage cargo release and background control subtracted. The lytic yeast peptide candidalysin (SIIGIIMGILGNIPQV IQIIMSIVKAFKGNK; Peptide Protein Research) was used as an additional positive control.

## Circular dichroism

The secondary structure of the human NINJ1 α-helix region (amino acids 40–69; HYASKKSAAESM LDIALLMANASQLKAVVE; GenScript Biotech Corporation) was measured using a Jasco J-1500 spectropolarimeter. For this assay, peptide solutions from a 10 mg/mL stock in 5% DMSO in ddH$_2$O were diluted to 0.2 mg/mL in buffer (20 mM sodium phosphate, pH 7.4, 10 mM NaCl, and 1% sodium dodecyl sulfate) with the final peptide concentration determined using a bicinchoninic acid (BCA) protein assay. NINJ1 helix peptide was titrated with glycine between 0 and 20 mM. 250 µL of the glycine titrated human NINJ1 helix peptide solutions were pipetted into 1 mm path length quartz CD cuvettes, capped, and placed in the Peltier temperature-controlled CD sample holder for analysis. A corresponding buffer blank spectrum was subtracted out from the sample datasets. In all conditions, spectra were recorded between 260 and 190 nm using 50 nm/min scanning speed, 0.5 nm data pitch, 1 s DIT, and 1 nm bandwidth. Between 3 and 20 accumulations were averaged. The CD studies were performed at the Structural and Biophysical Core Facility, The Hospital for Sick Children, Toronto, Canada.

## Quantification and statistics

TIRF and STED microscopy images of mouse macrophages were analyzed in Imaris (Oxford Instruments, Abingdon, United Kingdom) software. The Spot tool was used to identify NINJ1 puncta with parameters (estimated puncta *XY* diameter, region type, and quality filter) set using images from the LPS-primed group and applied to all samples across all experiments. Cells were contoured using the surfaces tool to measure cell area. The density of NINJ1 puncta was calculated on a per-cell basis by dividing the total number of puncta within the contoured plasma membrane area. TIRF microscopy

images of human primary macrophages were analyzed in FIJI, applying thresholds (triangle algorithm) and using the analyze particles tool to segment cells in the Cholera toxin subunit B channel. The find maxima command was used to identify NINJ1 puncta within these cell outlines, and the measurement tool was used to measure the area of the cell outlines. The density of NINJ1 puncta was calculated on a per-cell basis by dividing the number of puncta within a cell outline with its area.

Statistical testing was calculated using Prism 8.3.1 or 9.0 (GraphPad Software Inc, La Jolla, CA, USA). Experiments with more than two groups were tested using ANOVA with Tukey's multiple comparison test. Unless otherwise indicated, presented data are representative of at least three independent experiments or donors and are provided as mean ± SEM. All collected data was analyzed. Unless otherwise specified, for non-quantitative data, such as western blots, experiments were replicated three times with representative images shown in the figures.

### Animal and human studies

All animal studies were approved by the Hospital for Sick Children Animal Care Committee (AUP #47781). All human studies were conducted according to the principles expressed in the Helsinki Declaration and approved by the Regional Committee for Medical and Health Research Ethics (No. 2009/2245). Informed consent was obtained from all subjects prior to sample collection.

### Acknowledgements

We thank Dr Nobuhiko Kayagaki and Dr Vishva Dixit for providing the rabbit monoclonal anti-mouse NINJ1 antibody, Dr John Brumell for providing pneumolysin, and Anne Marstad for technical assistance. We thank Dr Greg Wasney at the Structural and Biophysical Core Facility, The Hospital for Sick Children, for his technical assistance with the circular dichroism studies. All imaging of the mouse and human cells was performed at the SickKids Imaging Facility at The Hospital for Sick Children and the Cellular and Molecular Imaging Core Facility at NTNU, respectively. Graphical abstract created with BioRender.com. The authors declare no competing financial interests.

This work was supported by a Mentored Research Award from the *International Anesthesia Research Society* and an Early Investigator Award from the Department of Anesthesiology and Pain Medicine, University of Toronto to BES; by grants (287696, 223255) from the Research Council of Norway to THF; a Ragon Early Independence Fellowship to CLE; NIH grants AI133524, AI093589, AI116550, and P30DK34854 to JCK; and a fellowship to PD by the Boehringer Ingelheim Fonds.

## Additional information

### Funding

| Funder | Grant reference number | Author |
|---|---|---|
| International Anesthesia Research Society | Mentored Research Award | Benjamin Ethan Steinberg |
| Department of Anesthesiology and Pain Medicine, University of Toronto | Early Investigator Award | Benjamin Ethan Steinberg |
| Research Council of Norway | 287696 | Trude Helen Flo |
| Ragon Institute of MGH, MIT and Harvard | Ragon Early Independence Fellowship | Charles L Evavold |
| National Institutes of Health | AI133524 | Jonathan C Kagan |
| Boehringer Ingelheim Fonds | PhD Fellowship | Pascal Devant |
| National Institutes of Health | P30DK3485 | Jonathan C Kagan |

| Funder | Grant reference number | Author |
|---|---|---|
| National Institutes of Health | AI116550 | Jonathan C Kagan |
| National Institutes of Health | AI093589 | Jonathan C Kagan |
| Research Council of Norway | 223255 | Trude Helen Flo |

The funders had no role in study design, data collection and interpretation, or the decision to submit the work for publication.

## Author contributions

Jazlyn P Borges, Formal analysis, Investigation, Writing – original draft, Writing – review and editing; Ragnhild SR Sætra, Allen Volchuk, Marit Bugge, Formal analysis, Validation, Investigation, Methodology, Writing – review and editing; Pascal Devant, Data curation, Formal analysis, Investigation, Methodology, Writing – review and editing; Bjørnar Sporsheim, Formal analysis, Investigation; Bridget R Kilburn, Formal analysis, Investigation, Methodology, Writing – review and editing; Charles L Evavold, Supervision, Methodology, Writing – review and editing; Jonathan C Kagan, Supervision, Funding acquisition, Writing – review and editing; Neil M Goldenberg, Formal analysis, Methodology, Writing – review and editing; Trude Helen Flo, Conceptualization, Data curation, Formal analysis, Supervision, Methodology, Writing – original draft, Project administration, Writing – review and editing; Benjamin Ethan Steinberg, Conceptualization, Data curation, Formal analysis, Supervision, Funding acquisition, Validation, Methodology, Writing – original draft, Project administration, Writing – review and editing

## Author ORCIDs

Ragnhild SR Sætra ⓘ http://orcid.org/0000-0002-8248-0460
Pascal Devant ⓘ http://orcid.org/0000-0001-9743-6764
Bridget R Kilburn ⓘ http://orcid.org/0000-0002-0171-9370
Charles L Evavold ⓘ http://orcid.org/0000-0003-3913-5626
Jonathan C Kagan ⓘ http://orcid.org/0000-0003-2364-2746
Neil M Goldenberg ⓘ http://orcid.org/0000-0003-2785-1852
Trude Helen Flo ⓘ http://orcid.org/0000-0002-2569-0381
Benjamin Ethan Steinberg ⓘ http://orcid.org/0000-0002-3070-0548

## Ethics

Human subjects: All human studies were conducted according to the principles expressed in the Helsinki Declaration and approved by the Regional Committee for Medical and Health Research Ethics (No. 2009/2245). Informed consent was obtained from all subjects prior to sample collection. All animal studies were approved by the Hospital for Sick Children Animal Care Committee (AUP #47781).

## Decision letter and Author response

Decision letter https://doi.org/10.7554/eLife.78609.sa1
Author response https://doi.org/10.7554/eLife.78609.sa2

# Additional files

## Supplementary files
• MDAR checklist

• Source data 1. This file contains the images of the full western blots with the relevant bands used in the figures highlighted in boxes.

• Source data 2. This file contains images of the full western blots used in the figures.

## Data availability

All data generated or analyzed during this study are included in the manuscript and supporting files, which includes the source data for the manuscript figures.

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

# Appendix 1

## Appendix 1—key resources table

| Reagent type (species) or resource | Designation | Source or reference | Identifiers | Additional information |
|---|---|---|---|---|
| Gene (*Mus musculus*) | Ninj1 | GenBank | ID: 18081 | |
| Gene (*Homo sapiens*) | Ninj1 | GenBank | ID: 4814 | |
| Strain and strain background (*Escherichia coli*) | Stellar Competent Cells | TarakaBio | Strain HST08 | |
| Genetic reagent (*Mus musculus*) | *Ninj1* knockout | **Kayagaki et al., 2021** | | C57Bl/6 strain background; Mixed-sex cohorts |
| Cell line (*Mus musculus*) | RAW-asc | InvivoGen | raw-asc | |
| Cell line (*Mus musculus*) | iBMDM TET3G | **Evavold et al., 2018** | | |
| Cell line (*Mus musculus*) | iBMDM inducible doxy-moNINJ1-FLAG | This paper | | Jonathan Kagan laboratory; described in the Methods section |
| Cell line (*Mus musculus*) | iBMDM inducible doxy-NT-GSDMD-BFP | **Evavold et al., 2021** | | |
| Cell line (*Homo sapiens*) | HeLa | ATCC | CCL-2 | |
| Cell line (*Homo sapiens*) | Platinum GP | Cell biolabs | RV-103 | |
| Transfected construct (human) | siRNA to NINJ1 | Thermo Scientific | HSS107188, HSS107190, HSS181529 | 20 nM |
| Transfected construct (mouse) | HA-NINJ1 | This paper | | Steinberg laboratory; described in the Methods section |
| Biological sample (*Mus musculus*) | Primary bone marrow-derived macrophages (wildtype BMDM) | This paper | | Freshly prepared from C57Bl/6 J mice; male and female donors; Steinberg laboratory |
| Biological sample (*Mus musculus*) | Primary bone marrow-derived macrophages (*Ninj1* KO BMDM) | This paper | | Freshly prepared from *Ninj1* knockout mice; male and female donors; Steinberg laboratory |
| Biological sample (*Homo sapiens*) | Primary human macrophages (human MDM) | This paper | | Female and male donors; information not disclosed; Flo laboratory |
| Biological sample (*Homo sapiens*) | Induced pluripotent stem-cell (iPSC) derived macrophages (iPSDM) | This paper | | iPSCs were obtained from European Bank for induced pluripotent Stem Cells; Male origin; Flo laboratory |
| Antibody | Anti-human NINJ1 (Mouse monoclonal) | R&D Systems | MAB5105, RRID: AB_11128852 | Immunofluorescence: 10 µg/mL; Western blot: 2 µg/mL |
| Antibody | Anti-mouse NINJ1 (Rabbit monoclonal) | Other; **Kayagaki et al., 2021** | | Gift from Dr Vishva Dixit and Dr Nobuhiko Kayagaki (Genentech, Inc, South San Francisco, USA); 10 µg/mL |
| Antibody | Anti-β-Actin (Rabbit monoclonal) | Cell Signaling Technology | 8457, RRID: AB_10950489 | Western blot: 1:1000 |
| Antibody | Anti-β-Actin (Mouse monoclonal) | Sigma | A5441; RRID: AB_476744 | Western blot 1:5000 |
| Antibody | Caspase-1 (Mouse monoclonal) | Adipogen | AG-20B-0042-C100; RRID: #AB_2755041 | Western blot 1:1000 |
| Antibody | Gasdermin D (Rabbit monoclonal) | Abcam | ab209845; RRID: #AB_2783550 | Western blot 1:1000 |
| Antibody | Anti-pro-IL1B (Rabbit polyclonal) | Genetex | GTX74034; RRID: #AB_378141 | Western blot 1:1000 |
| Antibody | Anti-HA (Rabbit monoclonal) | Cell Signaling Technology | 3724; RRID: AB_1549585 | Western blot 1:1000 |
| Antibody | HRP conjugated anti-mouse IgG (H+L) (Goat polyclonal) | Jackson ImmunoResearch | 115-035-003 | Western blot: 1:3000 |

*Appendix 1 Continued on next page*

*Appendix 1 Continued*

| Reagent type (species) or resource | Designation | Source or reference | Identifiers | Additional information |
|---|---|---|---|---|
| Antibody | HRP conjugated anti-rabbit IgG (H+L) (Goat polyclonal) | Jackson ImmunoResearch | 111–035003 | Western blot: 1:3000 |
| Antibody | Anti-GAPDH (Rabbit polyclonal) | Santa Cruz Biotechnology | sc-25778; RRID: AB_10167668 | Western blot 1:1000 |
| Antibody | Anti-GAPDH (Mouse monoclonal) | Cell Signaling Technology | 97166; RRID: AB_2756824 | Western blot: 1:1000 |
| Antibody | Anti-FLAG APC (Mouse monoclonal) | Abcam | Ab72569; RRID: AB_1310127 | Flow cytometry 1:100 |
| Antibody | Rabbit anti-Mouse IgG (H+L) Recombinant Secondary Antibody, Alexa Fluor 647 (Rabbit polyclonal) | Invitrogen | A27029, RRID: AB_2536092 | 2 µg/mL |
| Antibody | Cy3-conjugated AffiniPure Donkey Anti-Rabbit IgG (H+L) (Donkey polyclonal) | Jackson ImmunoResearch | 711-165-152 | STED: 1:100; TIRF 1:1000 |
| Antibody | Mouse IgG2a isotype control (Mouse polyclonal) | Serotec | MCA929F; RRID: AB_322271 | 10 µg/mL |
| Recombinant DNA reagent | pRETROX-TRE3G | TakaraBio | 631188 | |
| Recombinant DNA reagent | pRETROX-TRE3G-Flag-NINJ1 | This paper. | | Jonathan Kagan laboratory; described in the Methods section |
| Recombinant DNA reagent | pMSCV-Flag-NINJ1 IRES EGFP | This paper. | | Jonathan Kagan laboratory; described in the Methods section |
| Recombinant DNA reagent | pCMV-VSVG | Addgene | Addgene #8454 | |
| Recombinant DNA reagent | NINJ1 (Myc-DDK tagged) cDNA clone | Origene | MR225037 | |
| Recombinant DNA reagent | pcDNA3.1-HA | Addgene | Addgene #128034 | |
| Peptide, recombinant protein | Human NINJ1 residues 40–69 | GenScript BioTech Corp | | HYASKKSAA ESMLDIALLM ANASQLKAVVE |
| Peptide, recombinant protein | Sequence-scrambled analog of human NINJ1 (residues 40–69) | GenScript BioTech Corp | | IAAAAMKMY LANSLEHAK SLKVVLASQDS |
| Peptide, recombinant protein | Candidalysin | Peptide Protein Research | | SIIGIIMGILG NIPQVIQIIM SIVKAFKGNK |
| Peptide, recombinant protein | TNF | PeproTech | 300–01 A | 10 ng/mL |
| Peptide, recombinant protein | Pneumolysin | Other | | Gift from Dr John Brumell (The Hospital for Sick Children, Toronto, Canada) |
| Peptide, recombinant protein | IL-3 | PeptroTech | 200–03 | 25 ng/mL |
| Peptide, recombinant protein | M-CSF | PeproTech | 300–25 | MDM: 10 ng/mL iPSDM: 100 ng/mL |
| Peptide, recombinant protein | VEGF | R&D Systems | 293-VE-050 | 50 ng/mL |
| Peptide, recombinant protein | SCF | R&D Systems | 255-SC-050 | 20 ng/mL |
| Peptide, recombinant protein | BMP-4 | R&D Systems | 315 BP-010 | 50 ng/mL |
| Peptide, recombinant protein | Cholera toxin subunit B Alexa Fluor 488 conjugate | Invitrogen | C34775 | 1 µg/mL |
| Commercial assay or kit | CyQuant LDH cytotoxicity assay kit | ThermoFisher Scientific | C20301 | |
| Commercial assay or kit | Human IL-1β ELISA | R&D Systems | DY201 | |

*Appendix 1 Continued on next page*

*Appendix 1 Continued*

| Reagent type (species) or resource | Designation | Source or reference | Identifiers | Additional information |
|---|---|---|---|---|
| Commercial assay or kit | Mouse IL-1β ELISA | Invitrogen | 88–7013 | |
| Commercial assay or kit | CellTiterGlo Luminescent Cell Viability Assay | Promega | G7570 | |
| Chemical compound and drug | Glycine | Sigma | G7126 or 67126 (Sigma); 104201 (Millipore) | |
| Chemical compound and drug | FM1-43 | Invitrogen | T3163 | 5 µg/mL |
| Chemical compound and drug | FM4-64 | Invitrogen | T3166 | 5 µg/mL |
| Chemical compound and drug | CellTracker Deep Red Dye | Invitrogen | C34565 | 0.5 µM |
| Chemical compound and drug | Tetramethylrhodamine ethyl ester perchlorate (TMRE) | Sigma | 87917 | 10 nM |
| Chemical compound and drug | 1,2-dioleoyl-sn-glycero-3-phosphocholine (DOPC) | Avanti Polar Lipids | #850375 | |
| Chemical compound and drug | 1,2-dioleoyl-sn-glycero-3-phospho-l-serine (DOPS) | Avanti Polar Lipids | #840035 | |
| Chemical compound and drug | zVAD | InvivoGen | tlrl-vad | 25 µM |
| Chemical compound and drug | BV6 | InvivoGen | inh-bv6 | 10 µM |
| Chemical compound and drug | Nigericin | InvivoGen or Sigma | tlrl-nig (InvivoGen); N7146 (Sigma) | 20 µM |
| Chemical compound and drug | ABT-199 | Tocris | 6960 | 25 µM |
| Chemical compound and drug | Lipopolysaccharide from *E. coli* serotype 0111:B4 or serotype O55:B5 (LPS) | Sigma | LPS25; L6529 | 0.5 or 1 µg/mL |
| Chemical compound and drug | Y-27632 | StemCell Technologies | 72302 | 10 µM |
| Chemical compound and drug | Doxycycline | Sigma | D3447 | 2 µg/mL |
| Software and algorithm | Imaris | Oxford Instruments | RRID: SCR_007370 | |
| Software and algorithm | FIJI v2.0.0 | imagej.net/ software/ fiji | RRID: SCR_002285 | |
| Software and algorithm | CellProfiler v.4.2.4 | cellprofiler.org/ | RRID: SCR_007358 | |
| Software and algorithm | PRISM v8.3.1 or v9.0 | GraphPad Software Inc | RRID: SCR_002798 | |
| Other | ProLong Diamond Antifade Mountant | Invitrogen | P36961 | Mountant for immunofluorescnece studies as described in the Methods section |

