## [Editor Report]

It's been widely known that the amino acid Glycine can work as a cytoprotectant and inhibit cell death-associated plasma membrane rupture. However, a long-standing question has been: how does Glycine cytoprotection work? In this manuscript, the authors demonstrate that Glycine treatment inhibits the clustering of NINJ1 to preserve membrane integrity, which provides a significant advance in the cell death field.

---

## [Decision Letter]

**Decision letter after peer review:**

Thank you for submitting your article "Glycine targets NINJ1-mediated plasma membrane rupture to provide cytoprotection" for consideration by *eLife*. Your article has been reviewed by 3 peer reviewers, and the evaluation has been overseen by a Reviewing Editor and Nancy Carrasco as the Senior Editor. The reviewers have opted to remain anonymous.

Essential revisions:

All reviewers appreciate the importance of the topic. However, they all raise significant concerns with regards to the lack of mechanistic understanding provided on how Glycine inhibition of NINJ1-mediated plasma membrane rupture? This is of great significance and should be explored.

1 – Is there a direct interaction between Glycine and NINJ1?

2 – Can the authors provide evidence on the mechanism/s by which Glycine prevents NINJ1 oligomerization?

Please make note of the specific comments from the reviewers below.

*Reviewer #1 (Recommendations for the authors):*

What is the mechanism of Glycine inhibition of NINJ1-mediated plasma membrane rupture? Is the mechanism direct or indirect? To increase the impact of the work, the authors would have to show the direct binding of Glycine to NINJ1 and provide biochemical/ structural insights into how Glycine inhibits NINJ1. Direct binding can be measured by SPR analysis using full or partial length NINJ1 protein. Other amino acids that were previously shown to be cytoprotective (i.e. Serine) should be included in this SPR study as well. If inhibition by Glycine is actually indirect, further exploration of this mechanism would need to be provided.

*Reviewer #2 (Recommendations for the authors):*

Although the authors show that glycine inhibits the formation of large NINJ1 oligomers in cells, their study does not show any direct interaction of glycine with NINJ1 or shed light on the mechanism of NINJ1-mediated cell rupture.

(1) The authors initiated this study by assessing the functional and morphological similarities between NINJ1 knockout and glycine treatment during different types of cell death in both human and murine cells. This requires careful kinetics and concentration-dependent experiments, which this paper falls short of. Besides, the staining patterns with plasma membrane dye exhibited obvious differences between NINJ1 knockout and glycine treatment, although the cells in both conditions showed intact plasma membrane (Figure 1E).

(2) Although NINJ1 has been identified as the key mediator of plasma membrane rupture by forming oligomers on the cell membrane, it remains unknown how NINJ1 is activated and executes its membrane-rupturing function. The authors found glycine treatment blocks NINJ1 oligomerization, which indicates glycine inhibits the activation of NINJ1 during cell death. Thus, besides NINJ1 itself, upstream regulators or co-factors of NINJ1 may be targeted and inactivated upon glycine treatment. Dissecting how NINJ1 inhibition occurs in the presence of glycine would further substantiate the authors' hypothesis proposed in this study, and meanwhile may also delineate the mechanism of how NINJ1 is activated in response to cell death signals.

(3) Biophysical methods to demonstrate a direct in vitro interaction of NINJ1 with glycine, modification of NINJ1 by glycine, or effectiveness of glycine in minimizing leakage of liposomes containing NINJ1 would greatly add to the story and determine whether glycine's effect requires other molecules.

(4) The imaging used in the manuscript is not state-of-the-art and does not take advantage of higher resolution methods available that could be used to visualize NINj1 clustering on pyroptotic balloons and examine how it is altered by glycine.

(5) The native gels seem to indicate 2 NINJ1 bands in control cells that are not dying – the ~40 kD band mentioned a higher band (~200 KD) that might correspond to higher order multimers observed in the control cells.

(6) It would be useful to use necroptotic cells throughout as a negative control for glycine effects.

*Reviewer #3 (Recommendations for the authors):*

While the work in this manuscript may suggest a possible link between the cytoprotective function of glycine and NINJ1, in its current form the manuscript is largely correlative and does not provide definitive evidence to support the conclusion being made. Substantial experimentation is necessary to demonstrate the direct link between glycine treatment and the prevention of Ninj1 oligomerization.

1. Figure 1E – why are cellular contents nearly completely gone in the NINJ1 knockout cells treated with glycine? I would think that the NINJ1-knockout would have no phenotype in response to glycine treatment, as they both presumably go through the same pathway (ie the presence and absence of glycine in knockout cells should not matter for the morphology of the cells). Can the authors clarify this? The authors claim that the morphology of cells treated with glycine and cells genetically deficient in Ninj1 is the same but it does not seem that this is the case.

2. The use of microscopy, particularly TIRF, to investigate NINJ1 surface dynamics and oligomerization is nice. However, the microscopy in figures 1E, and 2B needs further quantification, as it is quite difficult to determine similarities or differences between panels. Quantification of the microscopy images would be important to allow this. In 2B in particular we see one panel and one close-up inset that basically shows a single cell for each condition, and no untreated condition is shown. This is not sufficient data to make a conclusion about the distribution of NINJ1 under these conditions.

3. What is the evidence that the cells remain viable with the morphology seen in Figure 1E in the presence of glycine or NINJ1 deficiency? Are the cells capable of carrying out biological functions in this state, or are the cells no longer metabolically active, but retain some level of membrane integrity and retain their cellular contents (like LDH and HMGB1)?

4. Prior studies demonstrate that NINJ1 does not contribute to cell lysis during necroptosis. Glycine is likewise not reported to play a role in protective responses after necroptosis. It is therefore surprising that this manuscript finds a role for NINJ1 and glycine in cytoprotection in the setting of necroptosis. This should be clarified and further investigated to account for the possible discrepancy if the authors wish to include necroptosis studies here.

5. The authors report that glycine limits NINJ1 oligomerization. What is the effect of glycine on GSDMD cleavage and oligomerization? In the absence of this data, it is not possible to definitively conclude that the effect of glycine is only on NINJ1.

6. The microscopy describing NINJ1 distribution in the presence of glycine is difficult to parse – it is unclear why the reduction in the number of NINJ1 puncta should be indicative of NINJ1 oligomerization. Again, images in Figure 3B need to be quantified, and it should also be made clear how the assay being quantified is linked to NINJ1 function. The authors quantify the density of puncta. It seems to me that the diameter or volume of the individual puncta should be increased as a relative indicator of aggregation. Is this possible to quantify? A control cell surface protein that would not be expected to behave this way under these conditions is critical in these experiments. In the absence of such a control it is not possible to conclude that the effect of LPS+nigericin on NINJ1 is specific, nor is it possible to conclude the effect of glycine on NINJ1 oligomerization is specific. In general negative controls for the specificity of glycine for NINJ1 in these assays is important to include in order to strengthen the conclusions being made here.

7. The authors are using the effect of glycine on NINJ1 to make the conclusion that the cytoprotective effect of glycine is mediated by interference with NINJ1 oligomerization. However, in their current form, the data are largely correlative. Definitive data to support this conclusion require demonstrating that forced oligomerization of NINJ1 in the presence of glycine restores the release of LDH and other cellular contents or a similar type of experiment that can definitively demonstrate that glycine limits cell lysis by directly regulating NINJ1 oligomerization.

8. It would be important to demonstrate in the authors' hands under the same conditions described here, what are the effect of glycine treatment on other parts of the pyroptosis pathway – casp1 cleavage, gsdmd processing and cleavage, pro-IL-1b levels, and IL-1b secretion.

[Editors’ note: further revisions were suggested prior to acceptance, as described below.]

Thank you for resubmitting your work entitled "Glycine targets NINJ1-mediated plasma membrane rupture to provide cytoprotection" for further consideration by *eLife*. Your revised article has been evaluated by Carla Rothlin (Senior Editor) and a Reviewing Editor.

The manuscript has been significantly improved but there are some remaining issues that need to be addressed before publication, as outlined below:

Please take note of the specific recommendations of Reviewers 1 and 2 below to revise your manuscript.

*Reviewer #1 (Recommendations for the authors):*

The submitted manuscript reveals that the amino acid, glycine, a known inhibitor of pyroptosis-related plasma membrane rupture, can block NINJ1-dependent plasma membrane rupture, either through direct or indirect mechanisms, perhaps at the level of NINJ1 clustering.

This manuscript confirms that glycine, a known weak inhibitor of pyroptosis-related cellular plasma membrane rupture (IC50 1.6-1.8 mM, PMID: 30975978), can inhibit NINJ1 clustering and consequent plasma membrane rupture.

New Supplement 3 data confirms that glycine does not inhibit pyroptosis (GSDMD-dependent cell death), which is consistent with previous observations in Ninj1-deficient cells (PMID: 33472215).

The mechanism of action for glycine inhibition, however, remains unclear from this manuscript; yet it is likely to be an indirect mechanism, as suggested from previous reports showing inhibitory activity of other amino acids (including alanine and serine (PMID: 30975978)).

Specific feedback

The authors made a herculean effort to revise the manuscript and diligently responded to reviewers’ feedback with new data. However, the mechanism of action through which glycine inhibits plasma membrane rupture is still not clear. Glycine inhibition could be mediated directly or indirectly through NINJ1.

I could recommend the publication if the authors revise the following points.

1. In the abstract, add the following or similar sentences.

Glycine prevents NINJ1 clustering either directly or indirectly.

Glycine does not prevent pyroptosis.

2. In the revised manuscript, include the alanine amino acid data (which is currently only used for rebuttal) and discuss the possible indirect mechanisms of action by glycine.

"its small size should generally allow free passage through the large membrane conduits involved in lytic cell death pathways, such as the gasdermin D pores formed during pyroptosis that are known to allow transport of small proteins and ions across lipid bilayers."

Does glycine need to be inside cells to block cell lysis? What is the evidence? This should be discussed with a citation. Amino acid transporter or another mechanism may contribute to the glycine intake.

*Reviewer #2 (Recommendations for the authors):*

The resubmission has addressed the reviewers' concerns and greatly improved the quality of the manuscript. Although the authors still have not identified the molecular basis of glycine inhibition of cell membrane bursting, they have now clearly shown that glycine only acts at the final step of Ninj1 clustering and membrane disruption, independently of the upstream steps of inflammasome activation and gasdermin pore formation.

However, their conclusion that Ninj1 is the target of glycine is not warranted by their data. In particular (1) their failed attempts to show a physical association of glycine with Ninj1 and (2) the lack of inhibition of Ninj1-mediated liposome leakage, but inhibition of Ninj1-mediated cell membrane rupture in cells, strongly argue that something else in the cellular membrane or associated with it (but not in the liposomes) is affected by glycine that then affects Ninj1 clustering.

It seems highly likely that identifying the target and mechanism behind glycine inhibition will be a very risky project with no easy solution. Therefore the experiments in this paper have gone about as far as they can and provide some very useful and well-controlled data for tackling this difficult project. I, therefore, think that the manuscript is suitable for publication with the current data. However, I feel strongly that the paper needs to be revised to

(1) discuss all the experimental methods that didn't show a physical association of glycine with Ninj1 (I don't think the negative data needs to be shown).

(2) remove the statements in the abstract and discussion that "Ninj1 is the target of glycine".

(3) emphasize that glycine does not inhibit lysosomal membrane perturbing properties of Ninj1 which suggests that a cellular Ninj1- and plasma membrane-associated molecule is likely the target of glycine.

(4) discuss any ideas about how glycine could be working (if you have them) and discuss what approaches could be used to identify the glycine target.

A more precise title would be a good idea (such as "Glycine inhibits NINJ1 membrane clustering to suppress plasma membrane rupture in cell death").

*Reviewer #3 (Recommendations for the authors):*

The authors have thoroughly and comprehensively addressed the comments, and the conclusions of the manuscript is significantly strengthened. The revised manuscript provides new insight into how glycine limits cell lysis and extends our understanding of the mechanisms of NINJ1-mediated cellular rupture. I have no further concerns or comments.

---

## [Author Response]

Essential revisions:All reviewers appreciate the importance of the topic. However, they all raise significant concerns with regards to the lack of mechanistic understanding provided on how Glycine inhibition of NINJ1-mediated plasma membrane rupture? This is of great significance and should be explored.1 – Is there a direct interaction between Glycine and NINJ1?2 – Can the authors provide evidence on the mechanism/s by which Glycine prevents NINJ1 oligomerization?

We acknowledge that our initial submission was limited as it did not offer a mechanistic understanding of how glycine inhibited NINJ1-mediated plasma membrane rupture. We demonstrated that glycine treatment prevents NINJ1 clustering as shown though biochemical and microscopy-based assays during activation of cell death pathways. As acknowledged in the Discussion section of our initial submission, this effect could be either indirect or direct action of glycine on NINJ1. In our revised manuscript we have undertaken multiple approaches to strengthen our findings and evaluate the mechanism of action of glycine on NINJ1.

We first turned our attention to whether glycine is able to interfere with NINJ1-mediated cellular rupture independent of upstream signaling pathways. To this end, we generated a doxycycline-inducible NINJ1 overexpression system in macrophages. Using this cell-based system, we demonstrate that glycine dosedependently suppresses NINJ1-mediated cell rupture independent of activation of upstream programmed cell death pathway activation (new Figure 5C-D). This new data is consistent with our finding that glycine does not prevent caspase-1 activation, gasdermin D processing or activation in pyroptosis (new Figure 5A-B). To further address glycine’s mode of action in this system, we evaluated its effect on NINJ1 clustering biochemically and found that glycine suppresses NINJ1 clustering in an over-expression system (new Figure 5E).

We next turned our attention to the amphipathic α-helix in the NINJ1 N-terminus (residues 40-69), which Kayagaki et al. demonstrated to be necessary for NINJ1-mediated plasma membrane rupture (e.g. Figure 4 of Kayagaki et al. Nature 2021). We evaluated whether glycine interferes with the membrane lytic function of the helix using a reconstituted liposomal rupture assay. In this system, we demonstrate that while the α-helix was able to rupture liposomes, glycine did not interfere with this effect (new Figure 5F).

A more comprehensive description of our approaches and new findings are as follows:

1. Inducible iBMDM-doxy-moNINJ1 system: In a new collaboration with Dr Jonathan Kagan’s group, we generated a doxycycline-inducible NINJ1 overexpression system in immortalized mouse macrophages. In this system, the overexpression of NINJ1 alone is sufficient to yield lytic cell death without the need for upstream activation mechanisms (e.g. LPS priming and nigericin). The cotreatment of these cells with glycine during induction of NINJ1 expression was able to suppress NINJ1-mediated plasma membrane rupture in a dose-dependent fashion. Notably, glycine did not change the frequency of NINJ1 expressing cells and only the highest concentration of glycine (50mM) slightly reduced NINJ1 surface expression. While these data do not answer whether glycine directly binds NINJ1 to prevent plasma membrane rupture, they further corroborate our findings that glycine is able to suppress NINJ1-mediated plasma membrane rupture and indicate that glycine cytoprotection is independent of upstream cell death pathway mediators. A characterization of the inducible iBMDM-doxy-moNINJ1 system is provided as Figure 5 —figure supplement 1.

– Methods details: iBMDMs with doxycycline-inducible expression of the N-terminal fragment of GSDMD fused to BFP are published (Evavold, Hafner-Bratkovic, Cell 2021). We used the same stock of iBMDMs from Cas9-expressing mice in which the TET3G transactivator protein had been introduced previously (Evavold et al., 2021) to generate cell lines with dox-inducible expression of NINJ1 carrying an N-terminal FLAG-tag. CDNA encoding full-length murine NINJ1 was ordered as a gBlock from Integrated DNA Technologies. This sequence was amplified by PCR to append an N-terminal FLAG-tag and cloned into pRETROX-TRE3G (TakaraBio) using BamHI and NotI restriction sites and into pMSCV IRES EGFP using NotI and SalI restriction sites. To introduce dox-inducible FLAG-NINJ1, 2.5 x 10^6^ Platinum-GP retrovirus packaging cells were seeded in a TC-treated 10 cm dish. On the next day, cells were transfected with 3 μg pCMV-VSV-G and 9 μg pRETROX-TRE3G encoding FLAG-NINJ1 using Lipofectamine 2000. After 18h, media was exchanged to 6 ml of fresh complete DMEM. The same day, 0.5 x 10^6^ Cas9- and TET3G-expressing iBMDMs were seeded per well in a 6-well plate. The next day, retroviral supernatant was harvested and filtered through 0.45 μm syringe filter, supplemented with polybrene (1:2000; EMD Millipore) and spun onto cells (1250 g, 30 °C, 1h). 6 ml of fresh cDMEM were added to retrovirus packaging cells and spinfection was repeated the next day. Three days after the second transduction, transduced cells were selected by growth in complete DMEM including 1.5 mg/ml G-418 and 10 μg/ml puromycin. To ensure maximal transgene expression, which correlates with the ability of FLAG-NINJ1 to induce cell death and lysis, FLAG-NINJ1 was also stably overexpressed in these cells using retroviral transduction with an MSCV retrovirus. Retrovirus production and spinfections were performed as described above, only using pMSCV IRES EGFP-FLAG-NINJ1 instead of pRETROX-TRE3G construct. Cells were then sorted on a BD Melody cell sorter for maximal EGFP expression (Top 5-10%). Clonal cell lines were derived by limited serial dilution. 1 x 10^5^ cells per well in a black 96-well plate in DMEM (Gibco)/10% FBS (R and D, S11550) were induced with 2 ug/ml doxycycline (Σ-Aldrich) in the presence or absence of increasing concentrations of glycine (Σ-Aldrich 67126) in a total volume of 200 μl for 8 hours (iBMDM GSDMD-NT) or 12 hours (iBMDM FLAG-NINJ1) before supernatants were harvested for LDH-assay (Invitrogen) and cells detached (PBS/4 mM EDTA) for flow cytometry (Becton Dickinson Fortessa). iBMDM GSDMD-NT express BFP and were analyzed directly; iBMDM NINJ1 were stained with an APC-labeled anti-FLAG antibody (1:100, Abcam ab72569) in PBS/2% FBS, 2.5 mM EDTA on ice and washed once before analysis.

2. Transient HA-mNINJ1 transfection system. In a complementary experiment to our inducible system, we performed transient, over-expression of HA-tagged mNINJ1 in HeLa cells, which have no detectable endogenous NINJ1. Analysis of NINJ1 clustering by native-PAGE revealed HAmNINJ1 aggregate at high molecular weight, which was prevented by 5 mM glycine treatment (new panel Figure 5E of our revised manuscript). This observation recapitulates the behaviour we observed in macrophages during pyroptosis, forms of necrosis, and post-apoptosis secondary necrosis.

– Method details: PCR reactions were performed using ConeAmp HiFi master mix (Takara). Primers for mouse NINJ1 (insert) PCR reaction were: Forward primer 5’gat tac gct gag tcg ggc act gag ga3’ and reverse primer 5’ctc ggt acc cta ctg ccg ggg cgc ca3’. Template DNA for the PCR reaction was mouse cDNA clone from Origene (Cat. No. MR225037). The template vector pcDNA3.1-HA was obtained from Addgene. Primers for pcDNA3.1-HA PCR (vector) reaction were: Forward primer 5’cgg cag tag ggt acc gag ctc gga3’ and reverse primer 5’gcc cga ctc agc gta atc tgg aac atc3’. PCR reaction products were cleaned up using the NucleoSpin PCR Clean-up kit (Macherey-Nagel). The PCR products contained 18bp overlapping regions between the pcDNA3.1-HA tag vector and the mouse NINJ1 insert. This allows for fusion of the insert and vector using the In-Fusion Snap Assembly EcoDry Cloning Kit (Takara, 638954) and transformation into Stellar competent bacterial cells (Takara). DNA sequencing with CMV-F primer confirmed NINJ1 insertion into the pcDNA3.1-HA vector and that NINJ1 was in-frame with the N-terminal HA-tag. HeLa cells were transfected with pcDNA 3.1HA-NINJ1 using Fugene HD transfection reagent (Promega) according to the manufacturer’s protocol.

3. In vitro liposomal rupture system. As reported by Kayagaki et al., the N terminus of NINJ1 contains a purported amphipathic helix between amino acids 40 to 69, which is necessary for NINJ1mediated plasma membrane rupture (Figure 4 of their paper in Nature 2021). Accordingly, we next turned our attention to this amphipathic helix as a potential target of glycine’s mode of action. To evaluate whether glycine functionally and directly interacts with this purported helix, we employed an in vitro large unilamellar vesicle (LUVs) liposomal system where we measure peptide-induced liposomal rupture using a fluorescence reporter. A similar system was used to demonstrate that the NINJ1 amphipathic helix is sufficient to rupture LUVs (Extended Data Figure 7C, Kayagaki et al. Nature 2021). Whereas the NINJ1α-helix leads to liposomal rupture, co-treatment with glycine (50 mM) did not suppress this effect, suggesting that glycine does not directly target the lytic function of this NINJ1 domain (new panel Figure 5F of our revised manuscript).

– Method details: To generate large unilamellar vesicles, 1,2-dioleoyl-sn-glycero-3phosphocholine (DOPC, Avanti Polar Lipids) and 1,2-dioleoyl-sn-glycero-3-phospho-lserine (sodium salt) (DOPS, Avanti Polar Lipids) were first prepared at 80% DOPC and

20% DOPS, freeze dried and hydrated with a solution containing 25 mM 6carboxyfluorescein. The suspension was bath sonicated, freeze-thawed and extruded (Avanti Mini Extruder) using a Nucleopore 0.1 μm membrane (Whatman) to yield large unilamellar vesicles. The liposomes were washed three times in PBS to remove remaining 6carboxyfluorescein not contained within the liposomes (spin at 13 000 rpm for 5 min). At this concentration, the fluorescence of 6-carboxyflurescein self-quenches inside the liposomes. With liposomal rupture, the 6-carboxyfluorescein is released into the bathing solution thereby releasing the self-quenching with fluorescence monitored in a spectrofluorometer (excitation 492 nm; emission 512 nm). The liposomal rupture assay was setup by mixing equal amounts of the prepared LUVs with 0.4 mg ml−1 human NINJ1 αhelix region peptide (HYASKKSAAESMLDIALLMANASQLKAVVE; GenScript Biotech Corporation) with either vehicle or glycine (50 mM) in PBS buffer. Where indicated, a sequence-scrambled analogue peptide (IAAAAMKMYLANSLEHAKSLKVVLASQDS; GenScript Biotech Corporation) was used as a negative control for the human NINJ1 α-helix region peptide. Following 45 minutes, all liposomes were then ruptured by adding Triton X100 and the maximum fluorescence (cargo release) recorded. The results were converted to percentage cargo release and background control subtracted. The lytic yeast peptide candidalysin was used as an additional positive control.

Additional biophysical methods to probe for a direct interaction between NINJ1 and glycine: While our liposomal rupture assay suggests that glycine does not interfere with the lytic function of the NINJ1 αhelix in an artificial liposomal system, it remains plausible that glycine still interacts with this domain or other components of the NINJ1 N-terminus to interfere with NINJ1 clustering, which would be consistent with our presented data. As a result, we attempted a variety of biophysical approaches to evaluate a direct interaction between the NINJ1 N-terminus (including the α-helix) and glycine.

4. Circular dichroism spectroscopy: We first interrogated the N-terminal α-helix by circular dichroism (CD) spectroscopy in the presence of glycine. Like the studies by Kayagaki et al., the CD spectrum of this region reveals an α-helical secondary structure with characteristic downward deflections at 208 nm and 222 nm. Glycine alone did not affect the CD spectrum of the helix, indicating that it did not alter the stability of the helical secondary structure.

– Methods details: The secondary structure of the human NINJ1 α-helix region (amino acids 40-69; HYASKKSAAESMLDIALLMANASQLKAVVE; GenScript Biotech Corporation) was measured using a Jasco J-1500 spectropolarimeter. For this assay, peptide solutions from a 10 mg/mL stock in 5% DMSO in ddH_2_O were diluted to 0.2 mg/mL in buffer (20 mM sodium phosphate, pH 7.4, 10 mM NaCl and 1% sodium dodecyl sulfate) with the final peptide concentration determined using a bicinchoninic acid (BCA) protein assay. NINJ1 helix peptide was titrated with glycine between 0 and 20 mM. 250 µL of the glycine titrated human NINJ1 helix peptide solutions were pipetted into 1 mm path length quartz CD cuvettes, capped, and placed in the Peltier temperature-controlled CD sample holder for analysis. A corresponding buffer blank spectrum was subtracted out from the sample datasets. In all conditions, spectra were recorded between 260-190 nm using 50 nm/min scanning speed, 0.5 nm data pitch, 1 second DIT, and 1 nm bandwidth. Between 3 and 20 accumulations were averaged.

5. Bio-layer interferometry and surface plasmon resonance: Our Reviewers suggested we evaluate a direct interaction between NINJ1 and glycine using surface plasmon resonance (SPR). To this end, we trialed bio-layer interferometry as a similar approach. In these experiments, we probed for an interaction between the NINJ1 α-helix and a histidine-tagged poly-glycine peptide made of a concatenated chain of 10 glycine residues. These experiments were confounded by non-specific binding of the NINJ1 α-helix to both Ni-NAD and anti-His antibody bio-layer interferometry probes. Therefore, the resulting binding data (not shown) were not interpretable. For completeness, we include a detailed description of the protocol we undertook to evaluate the glycine and NINJ1 interaction by bio-layer interferometry.

– Methods details: Either Ni-NTA biosensors (Sartorius Part # 18-5101) or anti-Penta-His biosensors (Sartorius Part # 18-5120) were used for each experiment in a 96-well plate (Greiner Part #655209) as required by the Octet RH16 Biolayer Inferometer (BLI) instrument manufacturer (Sartorius). Polyhistidine-tagged poly-glycine ligand

(HHHHHHGGGGGGGGGG, stock solution of 10 mg/mL, 3.13 mM, in 5% DMSO in ddH2O) was immobilized to the Ni-NTA or anti-Penta-His biosensors at 200 nM and then dipped into a titration of the human NINJ1 α-helix peptide (residues 40-69 as above; (GenScript Biotech Corporation)) analyte in assay buffer (20 mM sodium phosphate, pH 7.4, 10mM NaCl and 1% sodium dodecyl sulfate). Assay volume per well was 50 µL at a temperature of 25°C. The assay baseline drift was subtracted out using the protein-bound biosensor dipped into buffer instead of the respective analyte. A negative control experiment was included where the ligand was not loaded onto the biosensors and a titration of the same concentrations of analyte were tested to determine bare” biosensor binding profiles. A blocking step was added using Superblock (Thermo Fisher cat. #37515). The general BLI step conditions used were as follows: 1. Biosensor check in assay buffer (30 sec). 2. HisPolyG Peptide (ligand) immobilization (10 min). 3. Blocking with Superblock (3 min). 4. Baseline in assay buffer (3 min). 5. NINJ1 Helix Peptide analyte titration association (10 min). 6. Dissociation back into aseline buffer (10 min).

6. NINJ1 N-terminus and dynamic light scattering. As we did not see any evidence of a direct interaction between glycine and the α-helix of NINJ1 by CD or our in vitro liposomal assay, we next posited that glycine may interact with other regions of the NINJ1 N-terminus. This hypothesis would be consistent with our cellular data demonstrating that glycine interferes with NINJ1 clustering as residues purported to be involved in NINJ1 homotypic interactions may fall within the N terminus but outside the helical region (Bae SJ et al. J Cell Biochem 2017). To address this hypothesis, we attempted to purify the full N-terminus of mouse NINJ1 (residues 1 through 71) expressed in bacteria. In brief, His6-SUMO-mNINJ1 (residues 1-71) was made in a pET SUMO vector with TA cloning and expressed in BL21 Rosetta *E. coli*, prior to purification. Methodological details are provided below. The SUMO tag improved protein solubility. However, following removal of the His6-SUMO tag, the mNINJ1(1-71) aggregated into the insoluble pellet fraction across a variety of protein concentration, pH and salt concentrations. Its insolubility is consistent with NINJ1’s ability to aggregate / cluster within membranes during lytic cell death but precluded the use of the purified N-terminus in biochemical studies. As a mitigating strategy, we employed the His6-SUMOmNINJ1(1-71) protein in a series of dynamic light scattering experiments. In these experiments, we monitored for an effect of glycine on protein aggregation by measuring the extent of dispersion through varying glycine concentrations. While we observed a small effect of 5 mM glycine, the extent of protein dispersion was inconsistent across different glycine doses. We did not pursue these investigations given that they made use of the His6-SUMO-mNINJ1(1-71) protein.

– Methods details: Dynamic light scattering (DLS) experiments were conducted on a DynaPro Plate Reader III (Wyatt Technology). His6-SUMO-mNINJ1(1-71) protein, at the supplied stock concentration (2 mg/mL) and buffer, was titrated with varying concentrations of Glycine (40mM, 20mM, 10mM, 5mM, 1mM) and analyzed. Samples were transferred to a low volume 384-well microplate (Corning #3540) using 20 µL per well. The DLS experiment was conducted at 25°C with 10 acquisitions of 5 seconds, with laser autoattenuation enabled. Both intensity-weighted and number-weighted (right) regularization analyses were conducted.

Please make note of the specific comments from the reviewers below.Reviewer #1 (Recommendations for the authors):What is the mechanism of Glycine inhibition of NINJ1-mediated plasma membrane rupture? Is the mechanism direct or indirect? To increase the impact of the work, the authors would have to show the direct binding of Glycine to NINJ1 and provide biochemical/ structural insights into how Glycine inhibits NINJ1. Direct binding can be measured by SPR analysis using full or partial length NINJ1 protein. Other amino acids that were previously shown to be cytoprotective (i.e. Serine) should be included in this SPR study as well. If inhibition by Glycine is actually indirect, further exploration of this mechanism would need to be provided.

We now provide mechanistic studies into how glycine inhibits NINJ1-mediated plasma membrane rupture. A detailed description of these studies and new results are provided above. In brief, we demonstrate that glycine dose-dependently inhibits NINJ1-mediated rupture independent of action of upstream cell death pathways. Biophysical studies using circular dichroism demonstrated no evidence that glycine interacts with NINJ1 α-helix in its N-terminus (residues 40-69) consistent with our observation that glycine does not prevent its lytic function in an in vitro liposomal rupture assay. Together with our biochemical and imaging data, these new findings further support the notion that glycine interferes with NINJ1 clustering potentially through indirect interaction.

As our Reviewer also notes, other amino acids have been shown to be cytoprotective but with lower efficacy than glycine. In the context of pyroptosis, the extent of cytoprotection conferred by different amino acids and small molecules has been best characterized by Loomis WP et al. (Cell Death and Disease 2019). In their study, they demonstrate that glycine > alanine >> serine, valine can protect against membrane rupture during pyroptosis but did not implicate or study NINJ1. We have observed a similar pattern in our studies of NINJ1mediated membrane rupture. Representative data are shown here for LPS-primed primary mouse macrophages stimulated to undergo pyroptosis with nigericin without or with co-treatment with the indicated amino acid. These data are not included in our revised manuscript.

**Author response image 1. sa2fig1:** Glycine exhibits cytoprotection as compared to similar amino acisa through inhibition of NINK1 aggregation. Primary wildtype bone marrow-derived macrophages where induced to undergo pyroptosis in the presence of glycine, alanine, serine or valine at either 5 or 15 mM. (A) Similar to the work of Loomis et al. (Cell Death and Disease 2019), glycine exhibits a greater extent of cytoprotection compared with alanine >> serine, valine as measured by LDH release. (B) BN-PAGE for NINJ1 shows that amino acid treatment limits NINJ1 aggregation in pyroptossi with a similar hierarchy glycine > alanine > serine, valine.

Reviewer #2 (Recommendations for the authors):Although the authors show that glycine inhibits the formation of large NINJ1 oligomers in cells, their study does not show any direct interaction of glycine with NINJ1 or shed light on the mechanism of NINJ1-mediated cell rupture.(1) The authors initiated this study by assessing the functional and morphological similarities between NINJ1 knockout and glycine treatment during different types of cell death in both human and murine cells. This requires careful kinetics and concentration-dependent experiments, which this paper falls short of. Besides, the staining patterns with plasma membrane dye exhibited obvious differences between NINJ1 knockout and glycine treatment, although the cells in both conditions showed intact plasma membrane (Figure 1E).

We agree with the Reviewer that kinetics and dose-response of glycine inhibition of cell lysis would be informative. We have included time course data on morphology and cell viability (mitochondria integrity, ATP) in our response to Reviewer #1 Comment #5 above. As suggested by the Reviewer, we have now tested different concentrations of glycine and show dose-dependent inhibition of LDH release from human and mouse primary macrophages in response to LPS priming followed by nigericin, and from mouse iBMDMs in response to doxycycline-induced overexpression of NINJ1 or GSDMD-NT (see response to Essential Revisions above). These data are now included as supplemental material to Figures 1 and 2 in our revised manuscript.

(2) Although NINJ1 has been identified as the key mediator of plasma membrane rupture by forming oligomers on the cell membrane, it remains unknown how NINJ1 is activated and executes its membrane-rupturing function. The authors found glycine treatment blocks NINJ1 oligomerization, which indicates glycine inhibits the activation of NINJ1 during cell death. Thus, besides NINJ1 itself, upstream regulators or co-factors of NINJ1 may be targeted and inactivated upon glycine treatment. Dissecting how NINJ1 inhibition occurs in the presence of glycine would further substantiate the authors' hypothesis proposed in this study, and meanwhile may also delineate the mechanism of how NINJ1 is activated in response to cell death signals.

We agree that delineating the mechanism of NINJ1-mediated plasma membrane rupture is paramount to our understanding of lytic cell death pathways. As mentioned here, at present, it remains unknown how NINJ1 is activated and executes membrane rupture. Using multiple methods, we have demonstrated that glycine interferes with NINJ1 clustering. In the context of pyroptosis, we have previously shown that glycine does not interfere with IL-1β secretion (Volchuk A et al. Nature Communications 2020) in mouse macrophages, and we showed the same in Figure 2 —figure supplement 1 for human stem-cell derived macrophages (see Reviewer #1, Comment #1 above). In our revised manuscript, we expand on these findings by demonstrating that glycine treatment does not interfere with caspase-1 activation, gasdermin D cleavage, and IL-1β secretion. No specific proteins have been shown to link gasdermin D pore formation to NINJ1 activation. Therefore, we cannot evaluate them as potential glycine targets. However, using a doxycycline-inducible overexpression system of NINJ1 we provide new data showing that glycine targets NINJ1 independent of known upstream cell death pathways (see data in Response to Essential Revisions). Given the role of NINJ1 in mediating plasma membrane rupture in multiple lytic cell death pathways, a

generalized activation mechanism common to each of the implicated pathways is anticipated. As the mechanism of NINJ1 is explored and more fully delineated, we anticipate that additional molecular players and processes can be specifically evaluated as potential glycine targets.

(3) Biophysical methods to demonstrate a direct in vitro interaction of NINJ1 with glycine, modification of NINJ1 by glycine, or effectiveness of glycine in minimizing leakage of liposomes containing NINJ1 would greatly add to the story and determine whether glycine's effect requires other molecules.

We thank the reviewer for these helpful suggestions. In our revised submission, we now include (1) in vitro biophysical characterizations of glycine with the a-helix of NINJ1 using a liposomal rupture assay and circular dichroism spectroscopy, and (2) a doxycycline-inducible cellular system to isolate the effect of glycine on NINJ1 from other upstream molecules. These data and methods are described in full above.

(4) The imaging used in the manuscript is not state-of-the-art and does not take advantage of higher resolution methods available that could be used to visualize NINj1 clustering on pyroptotic balloons and examine how it is altered by glycine.

In our initial submission, we used total internal reflection fluorescence (TIRF) microscopy to limit our observations to the plasma membrane devoid of other intracellular compartments. TIRF microscopy is one of the few imaging modalities that allows for the fluorescence signal to be uniformly restricted to the cell’s plasma membrane.

Nevertheless, we acknowledge that TIRF is diffraction limited. As suggested by the Reviewer, we have advanced our imaging of native NINJ1 in macrophages undergoing pyroptosis without and with glycine treatment using stimulated emission depletion (STED) microscopy, a super-resolution fluorescence microscopy technique. These new data are provided in the updated Figure 3 of our revised manuscript. With our STED setup, we have improved our spatial resolution of 271 ± 98 nm with standard confocal microscopy to 63 ± 8 nm as measured by full-width at half maximum (FWHM) of single NINJ1 puncta. Using this approach in LPS-primed macrophages, we now better delineate the clustering of NINJ1 during pyroptosis, which is inhibited by glycine co-treatment. In brief, these data demonstrate that NINJ1 puncta decrease in

density upon pyroptosis stimulation with a corresponding increase in their mean intensity. Glycine treatment prevents this decrease in puncta density and decreased intensity. These new data, therefore, complement and corroborate the results presented in our initial manuscript using (1) native-PAGE, (2) cytotoxicity as measured by LDH release, and (3) TIRF microscopy. Together, our results support the notion that glycine targets NINJ1-mediated plasma membrane rupture and inhibits its clustering within the plasma membrane.

(5) The native gels seem to indicate 2 NINJ1 bands in control cells that are not dying – the ~40 kD band mentioned a higher band (~200 KD) that might correspond to higher order multimers observed in the control cells.

Thank you for this comment and astute observation. We agree that in the native gels of basal or LPS-primed cells, there is indeed a higher band of approximately 150-200 kDa. Of note, a similar band can be seen in the native gels shown in the paper by Kayagaki et al. (Nature 2021; see Figure 4C for example). As our Reviewer describes, this higher band may correspond to higher order multimers observed in the control cells. In our revised manuscript, we now mention this higher band and state that it may correspond to higher order multimers.

(6) It would be useful to use necroptotic cells throughout as a negative control for glycine effects.

We agree with the Reviewer that a negative control would strengthen our findings, and necroptosis is particularly interesting in this regard since previous studies have shown that glycine does not protect necroptotic cells from rupture, even though MLKL pores are formed (Chen X et al. Cell Research 2014). The study by Kayagaki N et al. further suggested that NINJ1 is not mediating PM rupture during necroptosis in mouse macrophages (although a partial phenotype was seen). We have now included data showing that glycine does not inhibit LDH release in necroptotic human primary macrophages (induced by TNF+zVAD+BV6). The new data are provided in the new Figure 2 —figure supplement 2 of our revised manuscript.

Reviewer #3 (Recommendations for the authors):While the work in this manuscript may suggest a possible link between the cytoprotective function of glycine and NINJ1, in its current form the manuscript is largely correlative and does not provide definitive evidence to support the conclusion being made. Substantial experimentation is necessary to demonstrate the direct link between glycine treatment and the prevention of Ninj1 oligomerization.1. Figure 1E – why are cellular contents nearly completely gone in the NINJ1 knockout cells treated with glycine? I would think that the NINJ1-knockout would have no phenotype in response to glycine treatment, as they both presumably go through the same pathway (ie the presence and absence of glycine in knockout cells should not matter for the morphology of the cells). Can the authors clarify this? The authors claim that the morphology of cells treated with glycine and cells genetically deficient in Ninj1 is the same but it does not seem that this is the case.

Thank you for this comment and notable observation. We agree that the presented images appear to depict significant differences between the glycine-treated wildtype and Ninj1 knockout cells. In the experiments of Figure 1E, we employ the FM fluorescent membrane dye to visualize the plasma membrane. In the context of pyroptosis stimulation without NINJ1 (or glycine treatment in wildtype cells), the membrane dye slowly enters the cytoplasm to eventually illuminate the intracellular membranes. We speculate that the dye is entering through the relatively large gasdermin D pores. The extent of intracellular fluorescent signal is a function of the time between pyroptosis induction and image acquisition. To avoid confusion, we have reviewed our images and have selected a representative set that were acquired at a more consistent time post-induction of pyroptosis.

2. The use of microscopy, particularly TIRF, to investigate NINJ1 surface dynamics and oligomerization is nice. However, the microscopy in figures 1E, and 2B needs further quantification, as it is quite difficult to determine similarities or differences between panels. Quantification of the microscopy images would be important to allow this. In 2B in particular we see one panel and one close-up inset that basically shows a single cell for each condition, and no untreated condition is shown. This is not sufficient data to make a conclusion about the distribution of NINJ1 under these conditions.

The figures presented in Figures 1E and 2B of our original submission are representative of the general morphology that we observed across multiple cells and independent experiments using wide-field microscopy (not TIRF). In these experiments, we have included LPS-primed but otherwise untreated cells as controls. We thank the reviewer for their positive acknowledgement of our investigation of NINJ1 surface dynamics and clustering by TIRF. As noted elsewhere, we have expanded the quantification of these data and now include additional corroborating super-resolution imaging by stimulated emission depletion (STED) of NINJ1 clustering in pyroptosis (Figures 3 and 4).

3. What is the evidence that the cells remain viable with the morphology seen in Figure 1E in the presence of glycine or NINJ1 deficiency? Are the cells capable of carrying out biological functions in this state, or are the cells no longer metabolically active, but retain some level of membrane integrity and retain their cellular contents (like LDH and HMGB1)?

We thank you for this is an important point. In the work of Kayagaki et al. (Nature 2021) that first positions NINJ1 in plasma membrane rupture, they show that Ninj1 knockout cells induced to undergo pyroptosis are in fact dead but not ruptured. In their paper, they state

“The cells were dead based on their loss of ATP, mitochondrial membrane potential, and motility (Figure 2g, Extended Data Figure 2h, Supplementary Videos 1, 2). Thus, PMR and related events, including LDH release and bubble disintegration, are genetically separable from GSDMD-driven cell death and IL-1β release. PMR is probably an event that occurs after cell death17,18. Of note, BMDMs ceased moving before bubble formation (Supplementary Videos 1, 2). NINJ1-independent loss of mitochondrial membrane potential also preceded PMR (as assessed by release of DD-150) (Extended Data Figure 2h, i).”

We have addressed these findings by evaluating mitochondrial membrane potential as well as ATP loss in human monocyte-derived macrophages stimulated to undergo pyroptosis with and without glycine. Glycinetreated cells lose both their mitochondria membrane potential (as measured by TMRE loss) and ATP comparably to untreated cells (data presented above in Reviewer #1, Comment #5). We (Volchuk et al. Nature Communications 2020) and others (Kayagaki N et al. Nature 2021) have previously shown that the glycine-treated pyroptotic macrophages retain large cellular contents, such as LDH and HMGB1.

4. Prior studies demonstrate that NINJ1 does not contribute to cell lysis during necroptosis. Glycine is likewise not reported to play a role in protective responses after necroptosis. It is therefore surprising that this manuscript finds a role for NINJ1 and glycine in cytoprotection in the setting of necroptosis. This should be clarified and further investigated to account for the possible discrepancy if the authors wish to include necroptosis studies here.

We agree with the Reviewer that the study by Kayagaki N et al. suggested that NINJ1 is not mediating PM rupture during necroptosis (although a partial phenotype was seen), and that previous studies have shown that glycine is not involved in cytoprotection during necroptosis (Chen X et al. Cell Research 2014). We believe this comment could result from a confusion around "necrosis" and "necroptosis". That said, we have now included data showing that glycine does not inhibit LDH release from necroptotic human primary macrophages (see Response #6 to Reviewer #2 above).

5. The authors report that glycine limits NINJ1 oligomerization. What is the effect of glycine on GSDMD cleavage and oligomerization? In the absence of this data, it is not possible to definitively conclude that the effect of glycine is only on NINJ1.

In our original submission, we did not include data that addressed whether glycine affects GSDMD cleavage or oligomerization, although we did show that glycine did not interfere with IL-1β secretion (Figure 4 Supplement 1). While our data demonstrate that glycine interfere with NINJ1 clustering and function in pyroptosis in addition to other forms of lytic cell death, we had not ruled out a potential effect of glycine on proteins upstream of NINJ1 in pyroptosis. We now include additional data that shows that glycine does not affect caspase-1 cleavage, gasdermin D cleavage, IL-1β processing and secretion. In these data, we use IL-1β secretion as a surrogate marker of gasdermin D oligomerization.

6. The microscopy describing NINJ1 distribution in the presence of glycine is difficult to parse – it is unclear why the reduction in the number of NINJ1 puncta should be indicative of NINJ1 oligomerization. Again, images in Figure 3B need to be quantified, and it should also be made clear how the assay being quantified is linked to NINJ1 function. The authors quantify the density of puncta. It seems to me that the diameter or volume of the individual puncta should be increased as a relative indicator of aggregation. Is this possible to quantify? A control cell surface protein that would not be expected to behave this way under these conditions is critical in these experiments. In the absence of such a control it is not possible to conclude that the effect of LPS+nigericin on NINJ1 is specific, nor is it possible to conclude the effect of glycine on NINJ1 oligomerization is specific. In general negative controls for the specificity of glycine for NINJ1 in these assays is important to include in order to strengthen the conclusions being made here.

We thank the Reviewer for these important points. We acknowledge that the size of the individual NINJ1 puncta could increase as a relative indicator of NINJ1 aggregation. Accordingly, in our revised submission, we now include additional analyses of NINJ1 puncta fluorescence intensity in our TIRF images of mouse and human macrophages (see Reviewer #1, Comment #7). This new analysis is complemented by super-resolution fluorescence microscopy to better resolve the size of the NINJ1 puncta. As noted above, we were able to improve our spatial resolution (mean ± SD) of 271 ± 98 nm with standard confocal microscopy to 63 ± 8 nm as measured by full-width at half maximum (FWHM) of single NINJ1 puncta. At this resolution, we still cannot fully resolve the membrane organization of these individual NINJ1 clusters. In addition, we provide Ab staining controls for the mouse and human TIRF imaging.

7. The authors are using the effect of glycine on NINJ1 to make the conclusion that the cytoprotective effect of glycine is mediated by interference with NINJ1 oligomerization. However, in their current form, the data are largely correlative. Definitive data to support this conclusion require demonstrating that forced oligomerization of NINJ1 in the presence of glycine restores the release of LDH and other cellular contents or a similar type of experiment that can definitively demonstrate that glycine limits cell lysis by directly regulating NINJ1 oligomerization.

In our revised submission we have strengthened our data that show that glycine interferes with NINJ1 oligomerization by both biochemical and imaging approaches in two species of macrophages. As described above, we now include informative biophysical studies evaluating a direct interaction of glycine with NINJ1.

While we do not have an engineered system that allows for the forced oligomerization of NINJ1, we now include data using an inducible NINJ1 system whereby the overexpression of NINJ1 alone leads to membrane rupture. These data – described in full above – show that glycine limits NINJ1-induced plasma membrane rupture independent of stimulation of any of pyroptosis, necrosis, or post-apoptosis lysis.

8. It would be important to demonstrate in the authors' hands under the same conditions described here, what are the effect of glycine treatment on other parts of the pyroptosis pathway – casp1 cleavage, gsdmd processing and cleavage, pro-IL-1b levels, and IL-1b secretion.

Thank you for this question. Please see our response to your Comment #5 above. In brief, glycine does not affect other parts of the pyroptosis pathway including caspase-1 cleavage, gasdermin D cleavage, or IL-1β processing and secretion. These data are included in our new Figure 5A-B of our revised manuscript.

[Editors’ note: further revisions were suggested prior to acceptance, as described below.]

Reviewer #1 (Recommendations for the authors):The submitted manuscript reveals that the amino acid, glycine, a known inhibitor of pyroptosis-related plasma membrane rupture, can block NINJ1-dependent plasma membrane rupture, either through direct or indirect mechanisms, perhaps at the level of NINJ1 clustering.This manuscript confirms that glycine, a known weak inhibitor of pyroptosis-related cellular plasma membrane rupture (IC50 1.6-1.8 mM, PMID: 30975978), can inhibit NINJ1 clustering and consequent plasma membrane rupture.New Supplement 3 data confirms that glycine does not inhibit pyroptosis (GSDMD-dependent cell death), which is consistent with previous observations in Ninj1-deficient cells (PMID: 33472215).The mechanism of action for glycine inhibition, however, remains unclear from this manuscript; yet it is likely to be an indirect mechanism, as suggested from previous reports showing inhibitory activity of other amino acids (including alanine and serine (PMID: 30975978)).Specific feedbackThe authors made a herculean effort to revise the manuscript and diligently responded to reviewers’ feedback with new data. However, the mechanism of action through which glycine inhibits plasma membrane rupture is still not clear. Glycine inhibition could be mediated directly or indirectly through NINJ1.

Thank you very much for your insightful comments regarding our manuscript throughout the review process. We acknowledge that the mechanism of action through which glycine inhibits NINJ1mediated plasma membrane rupture may be direct or indirect. This has been more fully discussed in our revised manuscript as outlined in the specific changes listed below.

I could recommend the publication if the authors revise the following points.1. In the abstract, add the following or similar sentences.Glycine prevents NINJ1 clustering either directly or indirectly.Glycine does not prevent pyroptosis.

We have added the first sentence to our abstract as suggested. For the second sentence (“Glycine does not prevent pyroptosis”), we have modified the abstract to include: “In pyroptosis, glycine preserves cellular integrity but does not affect upstream inflammasome activities or accompanying energetic cell death.” We feel this statement is consistent with our data that demonstrate that while glycine prevents pyroptotic cell membrane rupture, upstream inflammasome activities (e.g. IL-1β processing, GSDMD cleavage and pore formation; Figure 5) and accompanying energetic cell death (e.g. mitochondrial membrane depolarization, cellular ATP loss; Figure 2 —figure supplement 3) persist.

In addition, as suggested by our Reviewer #2 (Comment #2), we have also removed the statements “NINJ1 is the target of glycine…” from both the abstract and discussion. Together, these changes better emphasize that glycine is working at the level of NINJ1 through either direct or indirect mechanisms, consistent with our data.

2. In the revised manuscript, include the alanine amino acid data (which is currently only used for rebuttal) and discuss the possible indirect mechanisms of action by glycine.

Thank you for this suggestion. We now include our alanine (along with serine and valine) data in our revised manuscript in Figure 1 —figure supplement 1.

We have also expanded our discussion of possible indirect mechanisms of action of glycine in the Discussion section of our manuscript (see also Reviewer #2, Comments #3 and #4 below):

“The mechanism by which glycine interferes with NINJ1 clustering can be either direct or indirect. Our in vitro circular dichroism studies and liposomal rupture system respectively indicate that glycine does not interfere with the secondary structure or lytic function of the NINJ1 α-helix. It remains plausible that glycine still interacts directly with this domain or other components of the NINJ1 N-terminus in cells to interfere with NINJ1 clustering, consistent with our presented data. Alternatively, glycine may act on an unidentified intermediate that modulates NINJ1 clustering within the plasma membrane. Our data in pyroptotic cells suggest that such an intermediate would reside between active GSDMD and NINJ1. In an indirect model, any of a plasma membrane-associated protein, membrane lipid, or cellular metabolic pathway could be the target. In addition, it remains unclear whether glycine’s cytoprotective activity occurs on the intracellular or extracellular side of the plasma membrane. To the best of our knowledge, there is no direct evidence to suggest whether glycine’s cytoprotective effect is specifically mediated from the intracellular or extracellular side. Both extracellular (Weinberg et al., 2016) and intracellular (Rühl and Broz, 2022) sites of action have been proposed. Our in vitro data suggest that glycine is not acting directly on the N-terminal α-helix, which is thought to be extracellular; however, we do not otherwise delineate the topology of glycine’s mechanism of action.

Defining the site of action would provide insight into the molecular mechanism by which glycine inhibits NINJ1 clustering in the plasma membrane and ultimately help determine whether glycine directly or indirectly engages NINJ1. Indeed, how NINJ1 is activated by lytic cell death pathways, its clustering trigger, and method of plasma membrane disruption also remain important and outstanding questions in the field. Due to the ubiquity of lytic cell death pathways in human health and disease and the potency of glycine cytoprotection, answers to these questions will greatly advance the development of therapeutics against pathologic conditions associated with aberrant lytic cell death pathways.”

"its small size should generally allow free passage through the large membrane conduits involved in lytic cell death pathways, such as the gasdermin D pores formed during pyroptosis that are known to allow transport of small proteins and ions across lipid bilayers."Does glycine need to be inside cells to block cell lysis? What is the evidence? This should be discussed with a citation. Amino acid transporter or another mechanism may contribute to the glycine intake.

Thank you for alerting us to this important point for clarification. In our manuscript, we do not demonstrate that glycine is acting extra- or intracellularly. We have posited that given glycine’s small hydrodynamic radius (0.28 nm; doi.org/10.1021/j100799a040), it will freely enter a pyroptotic cell based on the structural studies of the gasdermin D pore size (PMID 33883744). Our intention in discussing the permeability of glycine in our *Introduction* was to emphasize that glycine’s mode of action is not through an osmoprotectant effect. Nevertheless, as noted, amino acid transporters or other mechanisms may indeed contribute to glycine intake.

The question of whether glycine is acting on the intracellular or extracellular side of the plasma membrane is intriguing. To the best of our knowledge, there is no direct evidence to suggest whether glycine’s cytoprotective effect is specifically mediated from the intracellular or extracellular side. Both extracellular (PMID 27066896) and intracellular (PMID 34537232) sites of action have been proposed. Our in vitro data suggest that glycine is not acting directly on the N-terminal a-helix, which is thought to be extracellular; however, we do not otherwise delineate the topology of glycine’s mechanism of action.

In our updated manuscript, we now discuss this question in our Introduction and as part of our Discussion of potential indirect and direct mechanisms.

Reviewer #2 (Recommendations for the authors):The resubmission has addressed the reviewers' concerns and greatly improved the quality of the manuscript. Although the authors still have not identified the molecular basis of glycine inhibition of cell membrane bursting, they have now clearly shown that glycine only acts at the final step of Ninj1 clustering and membrane disruption, independently of the upstream steps of inflammasome activation and gasdermin pore formation.However, their conclusion that Ninj1 is the target of glycine is not warranted by their data. In particular (1) their failed attempts to show a physical association of glycine with Ninj1 and (2) the lack of inhibition of Ninj1-mediated liposome leakage, but inhibition of Ninj1-mediated cell membrane rupture in cells, strongly argue that something else in the cellular membrane or associated with it (but not in the liposomes) is affected by glycine that then affects Ninj1 clustering.It seems highly likely that identifying the target and mechanism behind glycine inhibition will be a very risky project with no easy solution. Therefore the experiments in this paper have gone about as far as they can and provide some very useful and well-controlled data for tackling this difficult project. I, therefore, think that the manuscript is suitable for publication with the current data. However, I feel strongly that the paper needs to be revised to(1) Discuss all the experimental methods that didn't show a physical association of glycine with Ninj1 (I don't think the negative data needs to be shown).

Thank you for this suggestion, which buttresses our in vitro liposomal rupture data shown in Figure 5F. In our revised manuscript, we include a description and discussion of the experimental methods that did not show a physical association of glycine with NINJ1. Namely, circular dichroism (CD) of the N-terminal α-helix (residues 40-69) through a glycine titration from 0 to 20 mM did not affect the CD spectrum of the helix, indicating that it did not alter the stability of the helical secondary structure. These data are consistent with our findings that glycine does not prevent liposomal rupture induced by the α-helix (Figure 5F) and have been incorporated into a new Figure 5 —figure supplement 2.

We have not included the biolayer interferometry experiments in our revised manuscript because these experiments were confounded by non-specific binding of the NINJ1 α-helix to both Ni-NAD and anti-His antibody bio-layer interferometry probes. Therefore, the resulting binding data were not interpretable. Similarly, we have not incorporated our work with a His6-SUMO-tagged NINJ1 N terminus (residues 171) peptide. Unfortunately, following removal of the His6-SUMO tag, the mNINJ1(1-71) aggregated into the insoluble pellet fraction across a variety of protein concentrations, pH and salt concentrations. Its insolubility is consistent with NINJ1’s ability to aggregate / cluster within membranes during lytic cell death but precluded the use of the purified N-terminus in biochemical (or liposomal rupture) assays.

(2) Remove the statements in the abstract and discussion that "Ninj1 is the target of glycine".

As suggested, we have removed the statements “NINJ1 is the target of glycine…” from both the abstract and discussion.

(3) Emphasize that glycine does not inhibit lysosomal membrane perturbing properties of Ninj1 which suggests that a cellular Ninj1- and plasma membrane-associated molecule is likely the target of glycine.

Thank you for this important point. We have revised our Discussion section to emphasize that our finding that glycine does not inhibit liposomal membrane rupture by the N-terminal α-helix of NINJ1 is consistent with glycine targeting a plasma membrane-associated molecule or other non-NINJ1 molecular target. Because the α-helix of NINJ1’s N terminus is the minimally sufficient region for liposomal rupture, however, our in vitro findings do not exclude the possibility that other parts of full length NINJ1 are regulatory domains targeted by glycine. Accordingly, we now state that in cells glycine may act indirectly by targeting a plasma membrane-associated protein, the lipid membrane, or a metabolic pathway. A direct mechanism is still plausible through an interaction with a region of NINJ1 outside the N-terminal α-helix.

(4) Discuss any ideas about how glycine could be working (if you have them) and discuss what approaches could be used to identify the glycine target.

As noted above in our response # 3 (and Reviewer #1, Comment #3), we have clarified in our main text that glycine’s specific molecular target remains unknown and that both direct and indirect mechanisms remain possible. As suggested during our review, biophysical methods of testing for direct binding of glycine to either the full-length protein or the full N-terminus would provide important insight into potential direct mechanisms. Similarly, elucidating the mechanism by which NINJ1 is activated will most likely provide answers to how glycine works (and vice versa).

A more precise title would be a good idea (such as "Glycine inhibits NINJ1 membrane clustering to suppress plasma membrane rupture in cell death").

Thank you for this suggestion. We agree that “Glycine inhibits NINJ1 membrane clustering to suppress plasma membrane rupture in cell death” precisely encompasses our presented work. We have changed the title accordingly.